# Dogs fail to reciprocate the receipt of food from a human in a food-giving task

Jim McGetrick[1,2]*, Lisa Poncet[1,3,4], Marietta Amann[1], Johannes Schullern-Schrattenhofen[1], Leona Fux[1], Mayte Martínez[1], Friederike Range[1]

1 Domestication Lab, Department of Interdisciplinary Life Sciences, Konrad Lorenz Institute of Ethology, University of Veterinary Medicine, Vienna, Ernstbrunn, Austria, 2 Department for Farm Animals and Veterinary Public Health, Institute of Animal Welfare Science, University of Veterinary Medicine, Vienna, Austria, 3 Normandie Université, Unicaen, CNRS, EthoS (Éthologie animale et humaine), Caen, France, 4 Université de Rennes, CNRS, EthoS (Éthologie animale et humaine), Rennes, France

* Jim.McGetrick@vetmeduni.ac.at

**Data Availability Statement:** All relevant data are within the manuscript and its Supporting information files.

## Abstract

Domestic dogs have been shown to reciprocate help received from conspecifics in food-giving tasks. However, it is not yet known whether dogs also reciprocate help received from humans. Here, we investigated whether dogs reciprocate the receipt of food from humans. In an experience phase, subjects encountered a helpful human who provided them with food by activating a food dispenser, and an unhelpful human who did not provide them with food. Subjects later had the opportunity to return food to each human type, in a test phase, via the same mechanism. In addition, a free interaction session was conducted in which the subject was free to interact with its owner and with whichever human partner it had encountered on that day. Two studies were carried out, which differed in the complexity of the experience phase and the time lag between the experience phase and test phase. Subjects did not reciprocate the receipt of food in either study. Furthermore, no difference was observed in the duration subjects spent in proximity to, or the latency to approach, the two human partners. Although our results suggest that dogs do not reciprocate help received from humans, they also suggest that the dogs did not recognize the cooperative or uncooperative act of the humans during the experience phase. It is plausible that aspects of the experimental design hindered the emergence of any potential reciprocity. However, it is also possible that dogs are simply not prosocial towards humans in food-giving contexts.

## Introduction

Cooperation is observed in nature across a wide range of species and contexts [1]. It has recently been defined as the "simultaneous or consecutive acting together of two or more individuals by [the] same or different behaviours" [2]. One of the most prominent explanations for the evolution of cooperative behaviour, such as food sharing, between unrelated individuals is reciprocity (reciprocal altruism) [3–6]. Reciprocity generally refers to the reciprocal and contingent exchange of resources or services [3–6] (though exact definitions vary), including the exchange of different commodities [7–11]. Over the past number of decades, numerous studies

**Funding:** JM was funded by a DOC fellowship of the Austrian Academy of Sciences (ÖAW) (https://www.oeaw.ac.at/en/austrian-academy-of-sciences/) at the Konrad Lorenz Institute of Ethology, University of Veterinary Medicine, Vienna. JM and FR were funded by the Austrian Science Fund (FWF) (https://www.fwf.ac.at/de/) number W1262-B29. MM was funded by the Austrian Science Fund (FWF) number P30704 (https://www.fwf.ac.at/de/). The funders of this study did not play any role in the study design, data collection and analysis, decision to publish, or preparation of the manuscript.

**Competing interests:** The authors have declared that no competing interests exist.

have documented putative examples of reciprocity in non-human animal species [12–18]. Many aspects of reciprocity remain poorly understood, however, particularly the proximate mechanisms.

Domestic dogs are a promising model species for the study of reciprocity. Dogs have been shown to express helpful or prosocial behaviours (i.e. behaviours that benefit others [19, 20]), a prerequisite for reciprocity, towards conspecifics in experimental settings. For example, in a tray-pulling prosociality task, pet dogs pulled a tray to draw food towards a familiar conspecific in an adjacent enclosure significantly more times than they did in control conditions [21]. Likewise, in a token choice prosociality task, dogs chose the token that would result in a familiar conspecific in an adjacent enclosure being rewarded [22].

Dogs are also particularly interesting as they have been shown to express prosocial behaviours in interspecific interactions. Two recent experimental studies reported dogs directing prosocial behaviours towards humans. First, Carballo et al. [23] found that dogs opened the door of a box, in which their owner was trapped, more frequently when their owner expressed distress than when their owner remained calm. Moreover, they reported that such rescuing behaviour could not be accounted for by obedience, as subjects attempted to rescue their owner fewer times in a control condition in which the owner remained calm but called their dog by name. Similarly, in Van Bourg et al. [24], dogs rescued their owner from a box more often when they acted distressed than when they read aloud at the same level of volume as in the distressed condition. It is important to note, however, findings of prosociality directed to humans are inconsistent (see, for example, Bräuer et al. [25], Kaminski et al. [26], MacPherson and Roberts [27], Quervel-Chaumette et al. [28], and Sanford et al. [29]). Nonetheless, taken together, at least some of the studies outlined here indicate that dogs can behave prosocially towards both conspecifics and humans.

Further to the expression of prosocial behaviours towards conspecifics and humans, dogs have recently been shown to reciprocate help received from conspecifics. In a recent study conducted by Gfrerer and Taborsky [30], military dogs were exposed to a cooperative conspecific, who drew a platform containing food into their reach, and an uncooperative conspecific, who did not perform this behaviour. Subjects later had the opportunity to return the favour, both with previously cooperative and previously uncooperative partners and with partners to which they had not been exposed. Subjects pulled the tray to provide food to partners more frequently after experiencing the receipt of food in this setup. However, subjects did this for partners with which they had not been paired previously. This indicates that they helped any individual after the receipt of help, a pattern characteristic of a reciprocal strategy referred to as "generalized reciprocity" [31–33], rather than the classic form, now referred to as "direct reciprocity" whereby individuals directly help those that helped them before [3, 5, 34].

In a subsequent study, Gfrerer and Taborsky [35] employed two different food providing mechanisms to assess reciprocity in dogs. The first mechanism was the tray-pulling mechanism, as used in their previous study. The second was a Perspex box, containing food, which could be opened for a partner in an adjacent enclosure by pressing a lever. Importantly, the cooperative or uncooperative partner was supplied with only one of these mechanisms to provide the subject with food (or not) in the initial experience phase and the subject was later supplied with the alternative mechanism. Subjects successfully provided their former cooperative partner with food significantly more frequently than they did their former uncooperative partner, despite the fact that they were using a different mechanism. Thus, dogs successfully reciprocate help received from conspecifics in a relatively complex manner. However, to the best of our knowledge, it has not yet been investigated whether dogs reciprocate help received from humans.

Dogs possess additional characteristics which might suggest the propensity to reciprocate help received from humans. Apart from a long history of dog-human cooperation and communication [36–44] for which dogs appear to have evolved complex social cognitive traits [45, 46] (but see Udell et al. [47]), and the development of strong bonds with humans [48–52] which could facilitate reciprocity [53–55], dogs seem to distinguish between cooperative and uncooperative humans. For example, in a study investigating deceptive-like behaviour [56], dogs were exposed to a cooperative human who gave them a piece of food and an uncooperative human who did not give them food. They were later given the opportunity to lead the humans, separately, to three different locations, one containing the dog's preferred food, one containing a less desired food item, and one containing nothing. Subjects led the cooperative human to the location of the preferred food item significantly more often than the uncooperative human, indicating their ability to discriminate on the basis of cooperativeness.

Carballo et al. [57] also provided evidence for dogs' ability to discriminate between cooperative and uncooperative humans. Subjects observed two humans, in separate trials, pointing towards a bowl that contained a piece of food. The subject was allowed to approach the bowl and if the human was cooperative, he/she allowed the subject to eat the piece of food, whereas if the human was uncooperative, he/she quickly took the food and ate it, thereby preventing the subject from eating it. Although no differences in the latency to approach the two different human types was observed across an initial training block, in the second training block, subjects took significantly longer to approach the uncooperative human than the cooperative human. Furthermore, after the second block of training, in a choice test, subjects chose the cooperative human over the uncooperative human.

Several additional studies present similar findings, with dogs discriminating between a cooperative and uncooperative human [58–60]. At least two studies also demonstrate that dogs successfully discriminate between a cooperative and an uncooperative human after observation of third-party interactions, though alternative explanations such as dogs choosing based on the location of each human, or based on the behaviour of the cooperative or uncooperative human, rather than the interaction, mar these results [61–65]. Nevertheless, there is evidence that dogs can discriminate between a cooperative and an uncooperative human.

Given dogs' long history of cooperating with humans, their specialized skills for such interaction with humans, their capacity to develop bonds with humans, their ability to discriminate between a cooperative and an uncooperative human, and their propensity to reciprocate help received from conspecifics, we investigated whether pet dogs reciprocate help received from humans. To investigate potential reciprocation of help received, we applied a design generally matching that used to demonstrate reciprocity in dogs [30, 35] and rats [17]. Subjects were exposed separately to a helpful human, who provided them with food, and an unhelpful human, who did not provide them with food. In a later session, the dogs then had the opportunity to provide help to each of these humans. We additionally attempted to determine whether subjects discriminated between the helpful and unhelpful human, or developed a general preference for either human. Measures such as latency to approach and duration of proximity have previously revealed dogs' discrimination of individuals varying in cooperativeness [57, 66]. Accordingly, after the primary reciprocity tests, we conducted a free interaction session to discern a difference in subjects' latency to approach, and their duration of proximity to, each human.

## Study 1

### Methods

**Ethical approval.**   All procedures were approved by the ethics committee of the University of Veterinary Medicine, Vienna (ethical protocol no.'s: ETK-26/02/2019; ETK-134/07/2019).

Additionally, dog owners were required to sign a consent form prior to participation in the study.

**Subjects.** All dogs included in the study were pet dogs and were recruited via advertisements posted on social media and via word of mouth. The final sample tested in this study comprised 21 dogs (12 females; 9 males) of varying age (mean ± SD: 5.8 ± 2.8 years) and from a range of breeds, including mixed breeds (see Table 1). Eight additional dogs were unable to complete the training and were, therefore, not included in the study.

**Human partners.** Human participants, who acted as helpful or unhelpful partners in the setup (see below), were recruited through personal contact. They were briefly instructed on their task prior to participation. The final sample size of human participants was 12 (8 females; 4 males). The helpful and unhelpful partners were humans who were unfamiliar to the particular subject with which they were tested.

**General procedure and setup.** The study took place in a rectangular room (approx. 7 m x 6 m) and primarily within two adjacent, square enclosures (1.5 m x 1.5 m; see Fig 1), created by large-holed (15 cm x 10 cm) wire mesh fences with wooden frames (height = 1.1 m). One enclosure was reserved for the human (helpful or unhelpful) and the other enclosure was reserved for the study subjects.

The cooperative act performed by the helpful human comprised the provision of food. More specifically, the helpful human pressed a button to release dry dog food from a commercially available food dispenser (Trixie, Dog Activity Memory Trainer; cat no. 32040; TRIXIE Heimtierbedarf GmbH & Co. KG, Tarp, Germany). The unhelpful human pressed a non-functional button (i.e. a button that was identical to the functional button but that was switched off) and, therefore, failed to provide food. To provide food in return, dogs could press the same type of button, which released either small, round chocolates, chocolate covered almonds, or plain hazelnuts, depending on the human's preference (this was determined prior to the experiment by the experimenter).

The button, when switched on, emitted an audible sound when pressed and remotely controlled the food dispenser such that food was released. Regardless of whether a particular button was switched on or off, it was always presented in a green rubber holder which came with the product. The food dispenser could be adjusted so as to control the amount of food released each time the functional button was pressed. However, the amount of food could not be controlled precisely; thus, for dogs, the food dispenser released approximately five pieces of dry food each time the button was pressed and for humans the dispenser released approximately two pieces of food.

When the button was situated in the dog's enclosure, it was covered by a wooden box. The box was attached to a wooden base by hinges and springs. A rope was attached to the box so that the box could be opened from the experimenter's position by pulling. Only when the box was open, could the dog press the button. When the experimenter released pressure on the rope, the box closed again due to the force of the springs. A trial for the subject was defined as a single opening and closing of the box (i.e. a single presentation of the button).

Four cameras (one in each corner of the room) recorded general events that occurred in the room, while two webcams were used to record the activities within the two enclosures. Three outer fences of the enclosures were covered with black curtains so that the experimenter and owner could remain hidden from view during most phases of the experiment. The experimenter sat next to the human's enclosure and watched the proceedings via the two webcams. Dog owners could sit next to or behind the experimenter and could also watch the session on the experimenter's laptop, via the live feed from the webcams. Seven of the subjects were uncomfortable in the absence of their owner; therefore, the owner was permitted to sit behind the subject's enclosure, in view of the subject.

**Table 1. Subject information including age, sex, breed, and the number of training sessions required to reach the criterion.**

| Subject ID | Age (years) | Sex | Breed | No. of sessions to reach criterion |
|---|---|---|---|---|
| A1 | 5 | F | Australian shepherd | 1 |
| B1 | 10 | M | Border collie | 1 |
| C1 | 6 | M | Border collie | 1 |
| D1 | 11 | F | Border collie | 1 |
| E1 | 8 | F | Border collie | 1 |
| F1 | 7 | M | Greyhound | 3 |
| G1 | 5 | F | Terrier mix | 1 |
| H1 | 11 | M | Airedale terrier | 3 |
| I1 | 4 | M | Hungarian vizsla | 3 |
| J1 | 5 | M | Mix | 1 |
| K1 | 5 | F | Yorkshire terrier | 3 |
| L1 | 6 | M | Dachshund | 1 |
| M1 | 4 | M | Spanish galgo | 3 |
| N1 | 4 | M | Australian cattle dog mix | 1 |
| O1 | 2 | M | Labrador retriever mix | 1 |
| P1 | 1 | F | Border collie | 1 |
| Q1 | 7 | F | Podenco | 2 |
| R1 | 3 | F | Beagle | 2 |
| S1 | 3 | M | German hunting terrier mix | 2 |
| T1 | 5 | M | Australian shepherd mix | 1 |
| U1 | 9 | F | Australian shepherd | 2 |

F, female; M, male.

All testing took place at the Clever Dog Lab of the Messerli Research Institute, located at the University of Veterinary Medicine, Vienna.

**Training.** The training phase was conducted in five stages, as follows (see Fig 2):

Stage 1:

In the first stage of training, the button and dispenser were placed in the middle of the room, approximately half a metre apart. The subject was trained to press the button; the owner or experimenter attracted the subject's attention to the button verbally, by pointing, or by pressing the button. Each time the button was pressed, food rewards were released from the dispenser. The subject was permitted to retrieve these rewards. Once the subject had successfully pressed the button five consecutive times independently (i.e. without the assistance of the owner or the experimenter), it proceeded to the next stage of training.

Stage 2:

In stage two of the training, the button and dispenser were situated in the subject's test enclosure. The button remained in the box which was permanently open, and the food dispenser was positioned approximately half a metre away. Again, the dog was required to press the button and retrieve the food rewards from the dispenser five consecutive times to proceed to the next stage of training. The curtain covering one fence of the dog's enclosure was typically not lowered during this stage of training so that the dog could still see out into the room and see its owner.

Stage 3:

In stage three, movement of the box was introduced to separate trials. After each time the dog pressed the button, the box was closed for approximately four seconds and then

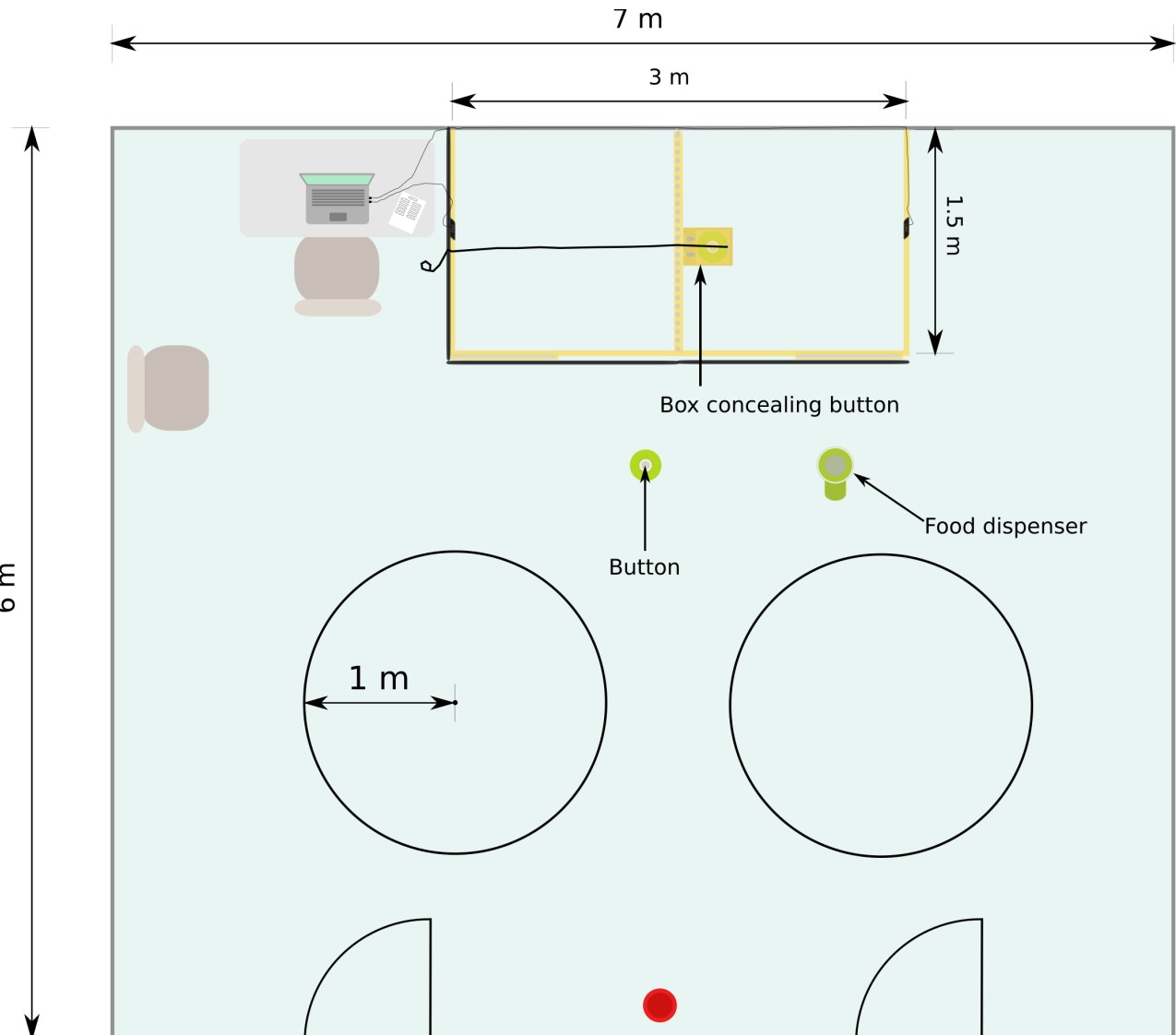

**Fig 1. Layout of test room.** Two circles (each approx. 1 m in radius) were marked on the floor of the room, for the free interaction session. A red water bowl was also present in the room. A food dispenser and button are depicted in the middle of the room (note: the dispenser and button were only in this position for the first stage of training). Black curtains surrounding some of the fences are represented by thick black lines.

reopened. The subject was required to press the button on five consecutive trials to progress to the next stage. Box opening and closing was continued for all subsequent stages. For dogs that were uncomfortable with the opening or closing of the box while eating, the dispenser was placed against the opposite fence. For one very tall dog, the dispenser was placed on an upturned box against the opposite fence so that it could eat from the food dispenser comfortably.

Stage 4:

In stage four, the food dispenser was placed in the human enclosure, directly adjacent to the box containing the button. A sliding door on the fence separating the two enclosures remained open. The subject was required to press the button in its own enclosure and then enter the neighbouring enclosure through the open sliding door, retrieve the food, and

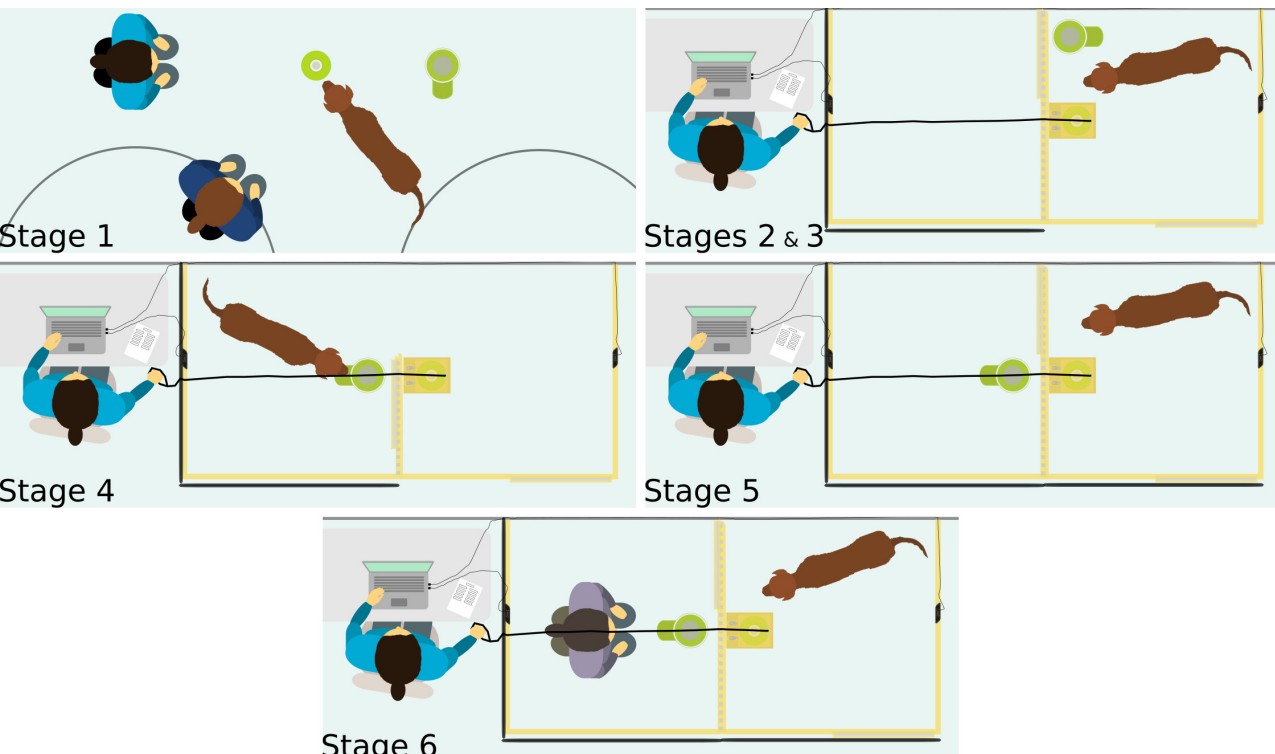

**Fig 2. Stages of training.** In stage 1, the owner and experimenter trained the subject to press the button to release food from the dispenser in the middle of the room. In stage 2, the subject was trained to press the button inside the permanently opened box in the enclosure to release food from the dispenser. In stage 3, opening and closing of the box was introduced (this was controlled by the experimenter pulling or releasing the rope, respectively). The subject could press the button each time the box was open to activate the dispenser. In stage 4, the dispenser was placed in the adjacent enclosure and a sliding door between enclosures was open. The subject could press the button each time the box was opened and then retrieve food from the dispenser in the adjacent enclosure. In stage 5, the sliding door between the enclosures was closed. The subject could press the button each time the box opened and food would be released from the dispenser; however, the subject could not access this food. In stage 6, the dispenser and a human were present in the adjacent enclosure. If the subject pressed the button when the box was opened, the human ate the food that was released from the dispenser. Note: in stage 6, typically a second person was not available; therefore, the experimenter played the role of the human and controlled the opening of the box while kneeling in the human enclosure.

return to press the button again. Subjects were permitted to proceed to the next stage of training once they had pressed the button and retrieved the food independently on ten consecutive trials.

Stage 5:

Stage five of the training was similar to stage four; however, the sliding door between the two enclosures was closed. Thus, if a subject pressed the button, food was released by the dispenser in the human's enclosure, but the subject could not enter the human's enclosure to retrieve it. Ten trials were conducted with this setup; however, the subjects were not required to press the button to proceed to the next stage. The purpose of this stage of training was to give subjects the opportunity to learn that, when the dispenser was on the other side of the fence and there was no opening, they could not retrieve the food. This stage of training matched the asocial control condition (see below), though the fence separating the enclosures in the asocial control condition did not have a sliding door. The curtain was typically lowered on one fence of the dog's enclosure in this stage of training.

Five motivational sessions were conducted after this stage of training. These motivational sessions were identical to stage three of training and were included to ensure that the button

pressing behaviour was not forgotten or extinguished by the previous experience of not receiving food.

Stage 6:

The final stage of training matched the previous stage; however, the human enclosure was occupied by a human who ate the food (human food) released from the food dispenser each time the subject pressed the button. The human eating the food in this stage was either the experimenter or a human unfamiliar to the dog; they were not one of the humans who would later act as the helpful or unhelpful human. Again, ten trials were conducted but the subject was not required to press the button in order to meet the criteria for inclusion in the study. The purpose of this stage of training was to give the subjects the opportunity to learn that, if the dispenser was in the human's enclosure, and a human was present, pressing the button resulted in the human being rewarded. Five motivational sessions were also conducted after this stage of training.

Between the stages of training, subjects were released from the enclosures and allowed to roam freely in the test room where they had access to their owner and water. If a subject showed signs of stress or lack of motivation, it was also given a break. However, if more than two breaks were required within a single stage of training, the session was terminated and continued on another day at that particular stage of training, preceded by a motivational session (provided the subject had already succeeded in stage three of training). Twelve subjects completed the training in a single session of one hour or less, four subjects required two separate sessions (i.e. two separate days), and five subjects required three sessions (i.e. three separate days).

**Experience phase.** Subjects received two experience phase sessions, one with the helpful human and one with the unhelpful human, each on a separate day (see S1 Video). The order in which the helpful and unhelpful humans were experienced was counterbalanced across subjects. Each experience phase consisted of five sessions conducted consecutively, with two-minute breaks in between. Each session consisted of ten trials.

Immediately prior to each experience phase, the human entered his/her enclosure before the dog entered the room so that the first encounter with the human was in this experimental setting. During the experience phase, the human knelt in the enclosure, facing the dog's enclosure (see Fig 3). Two buttons were available to the human regardless of whether he/she was playing the role of the helpful or unhelpful human. These buttons were adjacent to the dog's enclosure and were attached to either end of a rectangular board of wood such that one button was to the right of the human and one was to the left of the human if they were to kneel in the middle of the enclosure facing the dog's enclosure. One button was functional such that pressing this particular button resulted in the release of food rewards from the dispenser which was situated in the subject's enclosure. The food rewards used in the experience phase were dry food pieces. The type of dry food used in the experience phase for a subject was the same as that used in the training and motivational sessions for that subject. The other button was non-functional; pressing this button resulted in no food being released. The functional button was directly adjacent to the food dispenser. The food dispenser was, therefore, to the right or to the left of the dog when it faced the human's enclosure. The position of the dispenser and each button (i.e. functional and non-functional button) stayed the same for a particular dog but was counterbalanced across dogs.

The humans were instructed to press a button once every ten seconds, which was signaled by a timer hanging from the top of the fence separating the two enclosures, just above eye

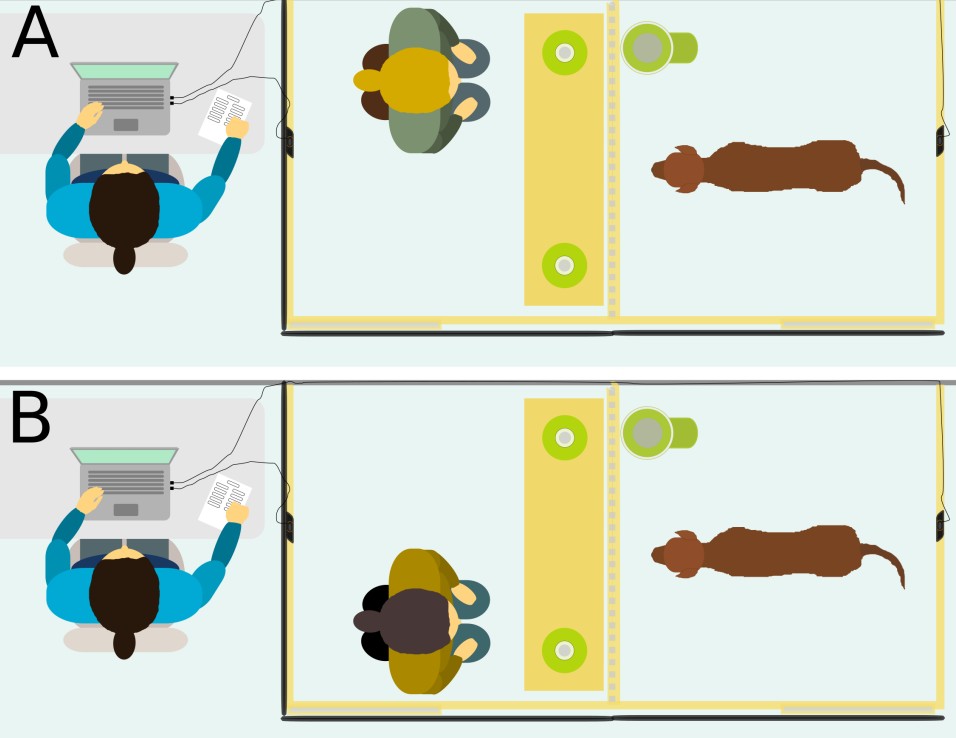

**Fig 3. Experience phase session with the helpful human (A) and with the unhelpful human (B).** With the exception of the first two trials of the experience phase, the helpful human knelt in front of the functional button, which was adjacent to the food dispenser, and pressed this button on each trial, thereby releasing food for the subject. With the exception of the first two trials, the unhelpful human knelt in front of the non-functional button and pressed this on each trial resulting in no food being released for the subject.

level. Humans were instructed not to talk to, and not to interact physically with, the subject. They were permitted to look at the subject but were asked not to stare at it.

For his/her first two trials, the helpful human knelt in front of the non-functional button and pressed this, before moving to the other, functional button, to press this on all remaining trials. The unhelpful human did the opposite, beginning with the functional button for the first two trials before switching to the non-functional button for all remaining trials. Each human began with the opposite button to which they were supposed to press, to facilitate the subject's understanding that the human intentionally provided or intentionally did not provide food. The unhelpful human was required to press a non-functional button to ensure that any difference in the subjects' propensity to press the button for the two humans in the test phase could not be due to differences in the activity level of the humans during the experience phase and could not be due to mimicking the behaviour of either one of the humans.

In between sessions, the subject was released from the enclosure for an approximately two-minute break in which they were free to roam around the room, interact with their owner or the experimenter, or drink from a bowl of water. The human remained in the enclosure during these breaks. After being released from the enclosure after each session, the subjects were provided with additional pieces of food in a food bowl. After sessions with the helpful human, they were provided with three to five pieces of food. However, after sessions with the unhelpful human, they received approximately 50 pieces of food to ensure that they received roughly the same amount of food independent of their experience with the human partner.

**Test phase.** The test phase consisted of two test days, one with the helpful human and one with the unhelpful human. The order in which subjects were tested with the helpful and unhelpful humans was counterbalanced across subjects and was independent of the order in which they experienced these two humans in the experience phase.

Each test day consisted of three conditions, four motivational sessions, and a free interaction session (described below). The three conditions included the test condition, the social facilitation control, and the asocial control (see Fig 4 and S1 Video). Each condition consisted of 20 trials. In each trial, the box was opened for approximately ten seconds, or until the subject pressed the button. The box remained closed for four seconds in between trials.

In the test condition, the human knelt in the middle of the human enclosure, facing the dog's enclosure (see Fig 4A). The food dispenser, containing food for the human, was positioned in front of the human against the fence. The button was located in the subject's enclosure. When the subject pressed the button, food was released from the food dispenser. The human took the pieces of food and ate them. If the dog attempted to press the button but failed, this was considered a successful trial and the human pretended to take a piece of food from the dispenser and ate a piece that they held in their hand from the previous trial.

The social facilitation control was similar to the test condition except that the food dispenser was positioned outside the subject's enclosure (see Fig 4B). The curtain on the outside fence of the subject's enclosure was removed for this condition so that the subject could see the food dispenser. Each time the button was pressed, food was released from the food dispenser but the human could not access it. The human remained passive in their enclosure. The social facilitation control was included to control for the possibility that button pressing was stimulated by the presence of a particular human, rather than the intention to provide the human with food. Such social facilitation effects [67] have been demonstrated with dogs [22] and chimpanzees [68] in similar experimental setups.

The asocial control matched the test condition except that the human was not present in the enclosure or in the room (see Fig 4C). The subject's pressing of the button still resulted in food being released from the food dispenser. The asocial control condition was included to control for the possibility that pressing the button in the test condition was simply a conditioned response or the result of subjects being generally motivated to press the button on a particular test day.

The order in which subjects experienced the three conditions was randomized across subjects. However, the order in which a particular subject experienced these three conditions was the same on both test days.

Each condition on the test day was preceded and succeeded by a motivational session in which the human partner left the room and the food dispenser was placed in the subject's enclosure for five trials such that the subject could reward itself five times. In total, there were four motivational sessions on each test day. There was an approximately one-minute break in between motivational sessions and experimental conditions. During these breaks, the experimenter changed the setup as necessary and the dog was free to explore the room and drink water.

**Free interaction session.** Each of the test days ended with a five-minute free interaction session in which the subject was free to roam around the room and interact with either the owner or the human with which it was tested on that particular day.

Two circles were marked with tape on the floor of the room, each with an approximately two-metre diameter, and with approximately one metre separating them (see Figs 1 and 5). The owner knelt or sat in the middle of one of the circles with their dog next to them. The experimenter left the room and the test human reentered the room and knelt or sat in the middle of the circle opposite the owner. The doors of the room were then closed and the owner

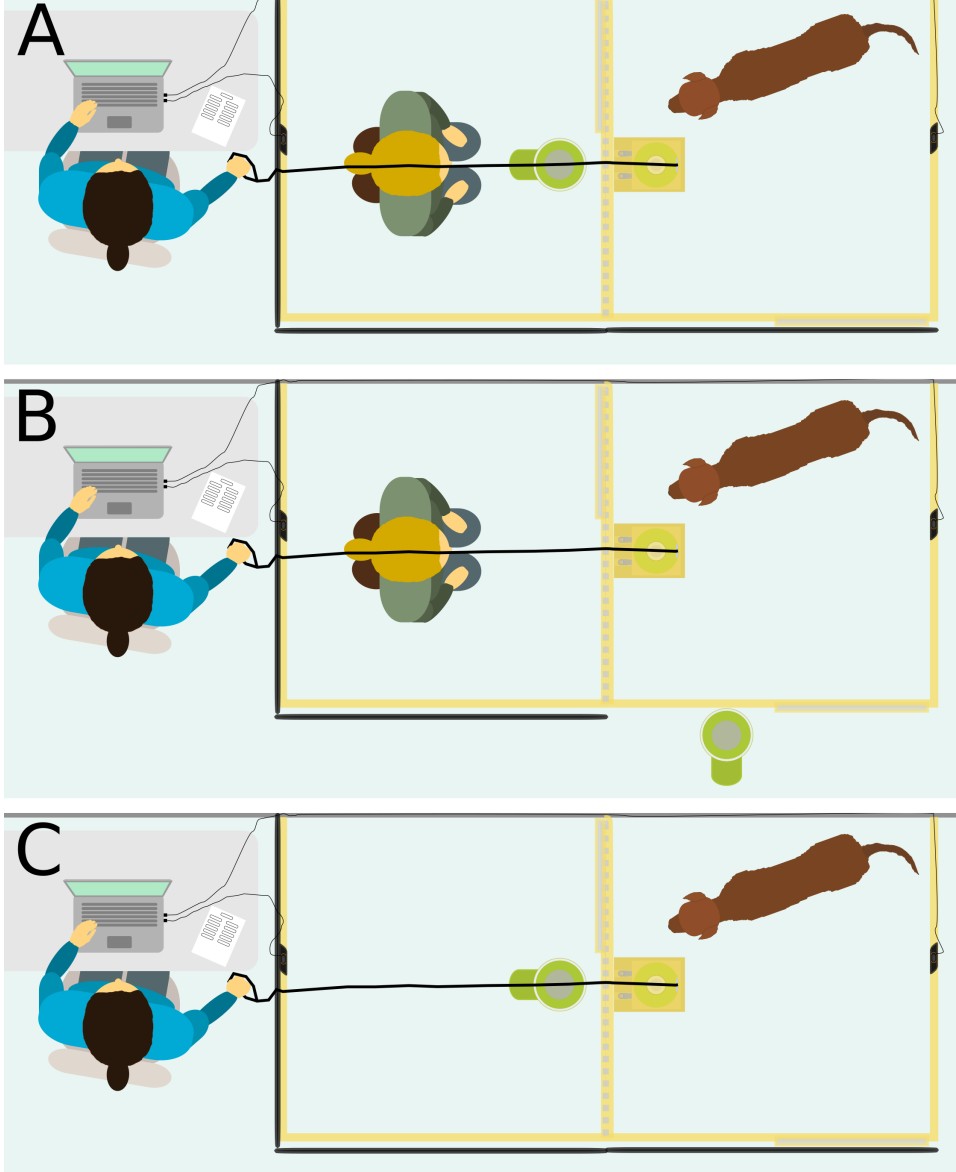

**Fig 4. Experimental conditions.** A, test condition; B, social facilitation control; C, asocial control. In the test condition, each time the subject pressed the button, food was released from the dispenser and the human ate this. In the social facilitation control, each time the subject pressed the button, food was released from the dispenser but the human could not access it. In the asocial control condition the human was not present. Each time the button was pressed in this condition, food was released but no human ate it.

released the subject. Both the owner and the test human were permitted to talk to each other casually during the five-minute session and both were permitted to pet the subject for five seconds each time the dog approached. They were not, however, permitted to call the subject or purposely attract its attention. For two subjects, the owner was unable to attend the test sessions so the experimenter took up the owner's position for these two subjects.

**Behaviour coding.** Behaviour coding was performed using Solomon coder (version beta 19.08.02 [69]). We coded the number of trials in which the subject pressed the button in each

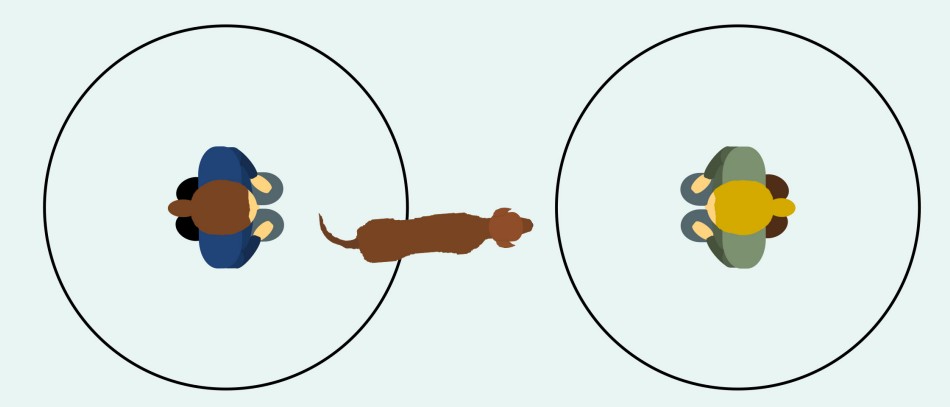

**Fig 5. Free interaction session.** The owner and the human partner with which the subject was tested on that day (helpful or unhelpful) knelt opposite each other in the middle of circles that were marked on the floor with tape (diameter = 2 m). The subject was released by its owner to roam free in the room, and to interact freely with the human partner or its owner, for 5 minutes.

test condition. Trials in which the subject attempted to press the button but failed either to press the button, or to activate the dispenser, were included (e.g. if the subject only touched the green rubber holder or if the button was pressed but not fully).

The latency to reach proximity to the test human and the duration spent in proximity to the test human in the free interaction session were also coded. A subject was considered in proximity to the human when at least one of its paws had crossed the line marking the circle.

A single video recording of one free interaction session for one subject was unavailable due to a technical malfunction. Therefore, this session could not be included in the final analysis. One person coded all the videos and a second experimenter coded approximately 20% of the test condition videos and free interaction session videos to assess interobserver reliability.

**Statistical analyses.** All models were fitted in R (version 3.6.2–4.0.2 [70]). Random slopes were identified, and overdispersion and model stability were assessed, using functions kindly provided by Roger Mundry (see below). Boxplots were created in ggplot2 (version 3.2.1–3.3.2 [71]) and cumulative incidence plots were created using the package "survminer" (version 0.4.6 [72]). All other packages and functions used are given below.

*Proportion of times the subjects pressed the button.* To analyse the effect of human type, condition, and their interaction, on the proportion of times subjects pressed the button, we fitted a Generalized Linear Mixed Model (GLMM) with a binomial error distribution and a logit link function [73]. In the model, we included the number of times subjects pressed the button and the number of times they did not press the button, as the response variable using the "cbind" function (see Baayen [74] for details). We included fixed effects of condition (i.e. test, social facilitation control, and asocial control), human type (helpful and unhelpful), and an interaction between condition and human type. These were the three terms of interest. To control for their potential effects, we also included test day order (i.e. whether it was the first or second test day), condition order (i.e. whether the specific conditions were first, second, or third on a particular test day), and first human experience (i.e. whether the subject experienced the helpful or unhelpful human in its first experience phase), as fixed effects.

We included random intercept effects of subject (i.e. the identity of the dog), human (the identity of the helpful or unhelpful human), the dog-human dyad, and observation (i.e. the identity of each individual observation; hereafter "observation level random effect"). The observation level random effect was included to account for session to session variation in the

propensity to press the button. In addition to random intercepts, to avoid overconfidence regarding the precision of estimates for the fixed effects, and to ensure that the type I error rate remained at the nominal level of 5%, we included all theoretically identifiable random slopes [75, 76]. We included random slopes of condition, human type, test day order, and condition order within the random effect of subject, we included random slopes of condition, human type, condition order, and first human experience within the random effect of human, and we included random slopes of condition and condition order within the random effect of the dog-human dyad. We also included the random slope of the interaction between condition and human type within the random effects of both subject and human. Condition, human type, and first human experience were manually dummy coded and centred for inclusion as random slopes. Prior to fitting the model, we z-transformed test day order and condition order to a mean of zero and a standard deviation of one to allow for an easier interpretation of results and to ease model convergence.

We fitted the model using the function "glmer" from the package "lme4" (version 1.1–23 [77]). The model could only be fitted after excluding the correlations among random intercepts and random slopes; therefore, we excluded these correlations to fit the model. As the random slope of the interaction between condition and human type within the random effect of subject was quite redundant with the observation level random effect, we fitted two additional models, one lacking the observation level random effect, and one lacking the random slope of the interaction between condition and human type within the random effect of subject, and we compared these models to determine whether the complete set of random effects was required. Based on log-likelihoods, which were essentially identical across the three models, and model complexity, for further inference we chose the model which lacked the random slope of the interaction between condition and human type within the random effect of subject (log-likelihoods: model with complete set of random effects: -305.7804 [$df$ = 30]; model lacking observation level random effect: -305.7804 [$df$ = 29]; model lacking the random slope of the interaction between condition and human type within the random effect of subject: -305.7804 [$df$ = 28]; see S1 Appendix for the final full model).

We confirmed that the best linear unbiased predictors (BLUPs) did not deviate from a normal distribution by visual inspection of histograms [74, 78]. We assessed collinearity by determining Variance Inflation Factors (VIFs) [79] using the function "vif" from the package "car" (version 3.0–8 [80]). This was applied to a model lacking the interaction between condition and human type, and random effects, and with the number of presses as the response variable. Collinearity did not appear to be an issue (maximum VIF: 1.021). The model was also not overdispersed (dispersion parameter = 0.697). To assess model stability at the level of the estimated coefficients and standard deviations, we excluded the levels of the random effects one at a time [81]. This revealed the model to be generally of acceptable stability with the possible exception of the intercept, the fixed effect of first human experience, the interaction between condition and human type, the random intercept effect of the dog-human dyad, the random slopes of condition and human type within the random effect of subject, the random slope of the interaction between condition and human type within the random effect of human, and the random intercept effect of human (see S1 Fig).

As an overall test of the effects of condition, human type, and their interaction, with the aim of avoiding cryptic multiple testing, we conducted a full-null model comparison [82], whereby the null model lacked these three terms but was otherwise identical to the full model. The full-null model comparison was based on a likelihood ratio test [83] using the R function "anova" and setting the "test" argument to "Chisq". The sample for this model included a total of 126 observations across 21 subjects, 12 humans, and 42 dyads.

To assess interobserver reliability, we calculated the intraclass correlation coefficient using the function "icc" in the package "irr" (version 0.84.1 [84]), setting the "model" argument to "twoway" and the "type" argument to "consistency". Interobserver reliability was excellent (ICC = 0.981, $n_{observations}$ = 25, $n_{raters}$ = 2, $P < 0.001$).

*Latency to approach the human.* To analyse differences in latency to approach the two different humans (helpful and unhelpful), we fitted a Cox proportional hazards regression model [85, 86]. We used the (start, stop] style of the model [87] and entered the response with a start time of zero for all subjects, the latency to reach proximity to the human, and status (i.e. whether they reached proximity to the human or not) using the "Surv" function in the package "survival" (version 3.1–8 [88, 89]). We included fixed effects of human type, test day order, and first human experience with human type being the fixed effect of interest.

As only one frailty term could be included in the model, we fitted two models, one with subject as the frailty term and one with human as the frailty term. We compared the log-likelihood and model complexity of the two models and selected the model with human as the frailty term for further inference (log-likelihoods: model with subject: -106.1143 [$df$ = 7.21]; model with human: -109.16779 [$df$ = 4.75]; see S1 Appendix for final full model).

We fitted the model using the function "coxph" in the package "survival" (version 3.1–8 [88, 89]). Prior to fitting the model, we z-transformed test day order to a mean of zero and a standard deviation of one to allow for an easier interpretation of results.

We tested the assumption of proportional hazards using the "cox.zph" function from the package "survival" (version 3.1–8 [88, 89]) and by visual inspection of scaled Schoenfeld residuals using the "ggcoxzph" function in the package "survminer" (version 0.4.7 [90]; see S2 Fig). Based on assessment of both, the assumption of proportional hazards was upheld. We assessed model stability by visual inspection of plots of dfbeta residuals using the function "ggcoxdiagnostics" from the package "survminer" (version 0.4.6 [72]; see S3 Fig). Model stability was deemed acceptable.

As an overall test of the effect of human type, we conducted a full-null model comparison [82], as above. The sample for this model included a total of 41 observations across 21 subjects and 12 humans. Interobserver reliability for latency to approach the test human was excellent (ICC = 1, $n_{observations}$ = 8, $n_{raters}$ = 2, $P < 0.001$).

*Duration of proximity to the human.* To determine whether human type had an influence on the proportion of time subjects spent in proximity to the human in the interaction session, we fitted a GLMM with a beta error distribution and a logit link function [91–93]. As such beta models cannot handle zeroes and ones, and our dataset comprised at least one zero, we transformed the response using the following formula (where "x" is the variable to be transformed; see Smithson and Verkuilen [93]):

$$x' = \frac{x.(length(x) - 1) + 0.5}{length(x)}$$

We included the fixed effect of human type as the primary term of interest. To control for their possible effects, we also included test day order and first human experience as fixed effects. We included random intercept effects of subject and human. No random slopes were identifiable for this model; therefore, no random slopes were included.

We fitted the model using the function "glmmTMB" from the package "glmmTMB" (version 1.0.0 [94]). Prior to fitting the model, we z-transformed test day order to a mean of zero and a standard deviation of one to allow for an easier interpretation of results and to ease model convergence. The full model can be seen in S1 Appendix.

The model was not overdispersed (dispersion parameter: 1.034). We determined VIFs [79] by applying the function "vif" from the package "car" (version 3.0–6 [80]) to a linear model. Collinearity did not appear to be an issue (maximum VIF: 1.032). The model's stability was generally acceptable (see S4 Fig).

A full-null model comparison was carried out, as above, as an overall test of the effect of human type. The sample for this model included a total of 41 observations across 21 subjects and 12 humans. Interobserver reliability for duration of proximity to the test human was excellent (ICC = 0.979, $n_{observations}$ = 8, $n_{raters}$ = 2, $P < 0.001$).

## Results

**Proportion of times the subjects pressed the button.**   Overall, neither condition, human type, nor the interaction between condition and human type had a significant effect on the proportion of times the subjects pressed the button (full-null model comparison: $\chi^2$ = 4.388, $df$ = 5, $P$ = 0.495; Fig 6).

**Latency to approach the human.**   There was no significant difference in the latency to approach the human in the interaction session based on whether they had been helpful or unhelpful (full-null model comparison: $\chi^2$ = 0.324, $df$ = 0.79, $P$ = 0.477; Fig 7).

**Duration of proximity to the human.**   Human type did not have a significant effect on the proportion of time subjects spent in proximity to the human (full-null model comparison: $\chi^2$ = 0.035, $df$ = 1, $P$ = 0.851; Fig 8).

## Discussion

Dogs did not reciprocate the help received from the humans in this study. There are two aspects of the setup which might have hindered the emergence of any reciprocity. First, there was a considerable time lag between the experience phase and the test phase for most subjects. For example, the number of days from the last experience phase session to the first test phase session ranged from 1 to 22 with a mean (± SD) of 8.5 (± 4.5) days and a median of 7 days. This could have placed large demands on the subjects' memory, a cognitive capacity which has been identified by other authors as a major factor constraining the emergence of reciprocity [95, 96]. Dogs have previously been shown to remember learned discrimination of images for at least 6 months [97]; thus, one might not expect memory to be an issue in our study, but initial exposure to the two human partners here was brief, potentially reducing the likelihood that they would be remembered for very long.

Second, the procedure during the experience phase, with both humans pressing a button, may have introduced too much complexity such that it confused the subjects or at least reduced the likelihood of them interpreting the actions of both humans accurately. We, therefore, conducted a second study to address these potential issues.

## Study 2

Study 2 largely followed the design and approach of study 1. There were, however, two major differences introduced in order to simplify the setup.

The first difference relates to the experience phase. During the experience phase in study 2, the human was presented with only one button, in contrast to study 1 in which the human was presented with two buttons. The human knelt in the middle of the human enclosure, facing the dogs' enclosure. The button was presented to the human in a box, as it was presented to subjects in study 1. The experience phase consisted of two sessions each of ten trials (a trial being defined by the opening and closing of the box). On each trial, the experimenter opened the box for five seconds. In the experience phase with the helpful human, the human pressed

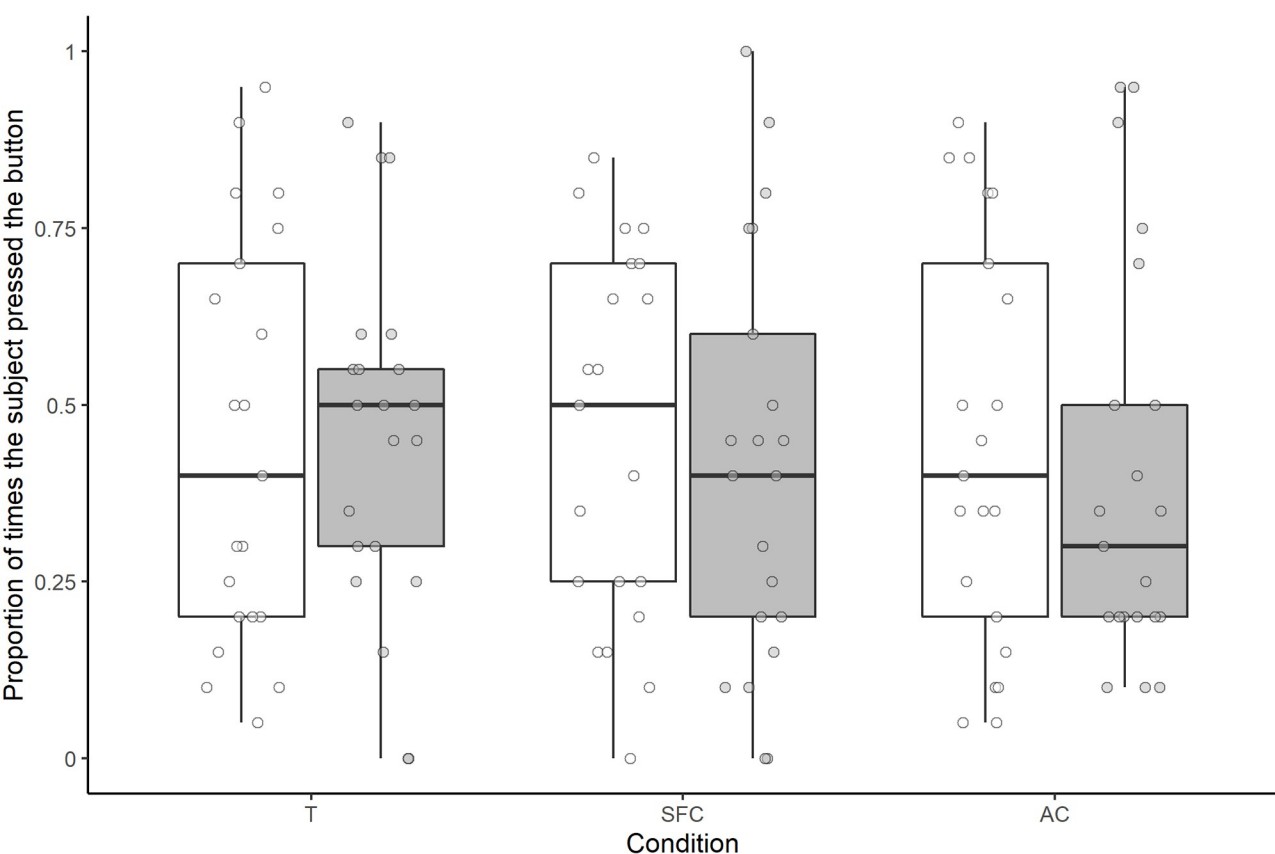

**Fig 6. Proportion of times the subjects pressed the button in each condition for each human type.** Pressing for the helpful human is represented in white and pressing for the unhelpful human is represented in grey. Boxes display the interquartile range, black horizontal bars represent the median, whiskers represent the range of data points within 1.5 times the interquartile range from the upper and lower hinge, and circles represent individual data points. T, test; SFC, social facilitation control; AC, asocial control.

the button after three seconds of the box opening. This resulted in food being released from the dispenser for the subject. In experience phase sessions with the unhelpful human, the button was not pressed upon opening of the box. Therefore, the subject did not receive any food from the food dispenser. As in study 1, the experience phase with each human type was conducted on a separate day and the order in which the two human types were experienced was counterbalanced across subjects.

In between the two sessions of the experience phase with each human, there was an approximately two-minute break in which the subject was released from the enclosure and was free to interact with its owner, the experimenter, or to drink water. Subjects were not provided with food in between experience phase sessions; thus, the amount of food received in each experience phase session differed.

The second major difference in the design of study 2 is that subjects were given the opportunity to reciprocate on the same day that they had the experience phase with a particular human type. The test conditions proceeded exactly as described for the test day in study 1, with a motivational session preceding and succeeding each condition. Other differences between the studies are identified below. In between the experience phase and the test phase, subjects were given an approximately two-minute break while the setup was changed. Subjects were free to roam around the room and drink water in this time.

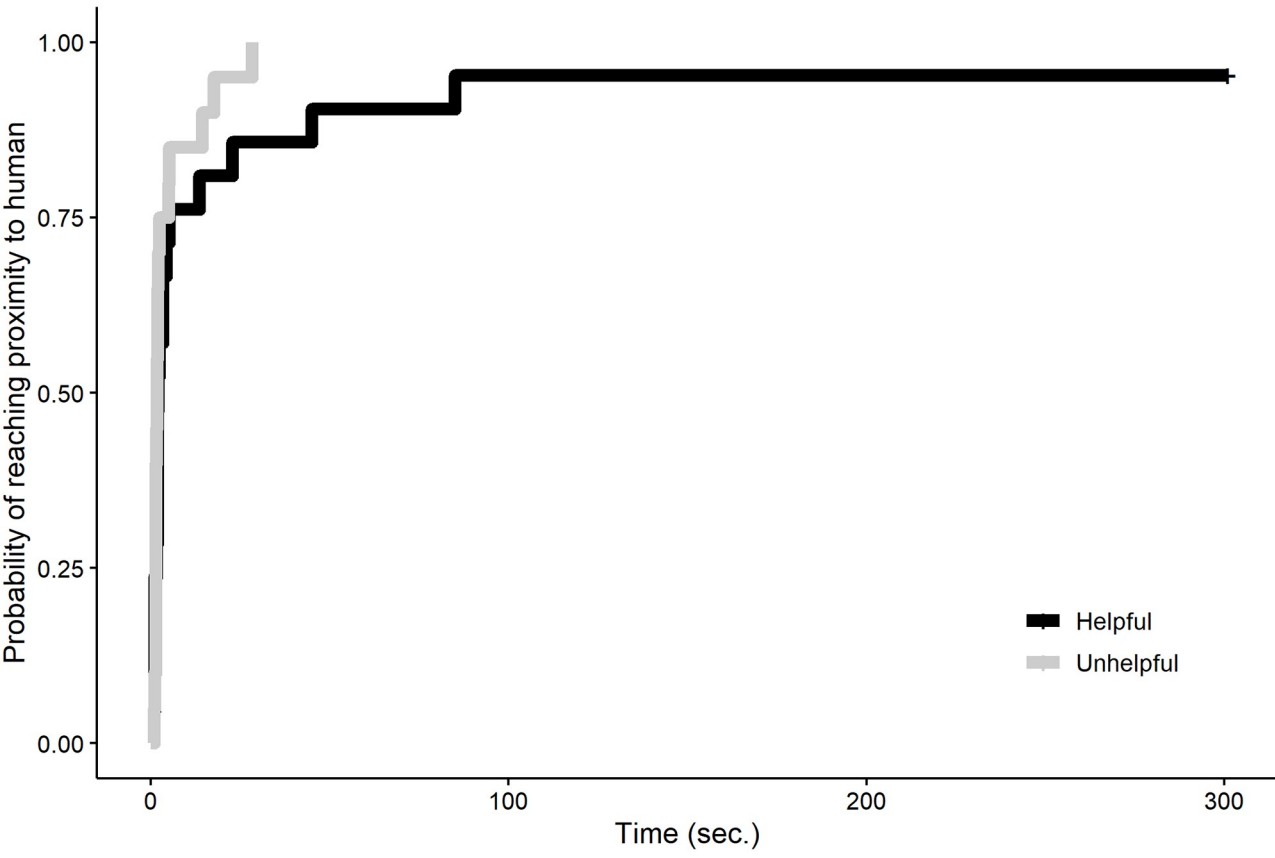

**Fig 7. Probability of reaching proximity to the human over time in the free interaction session.**

### Subjects

This study also tested 21 subjects (11 females; 10 males) from a range of breeds, including mixed breeds (see Table 2). The mean age (± SD) of this sample was 6.80 (± 2.74) years. Five of these subjects had taken part in study 1. Twenty-four female humans participated as the help-ful or unhelpful humans. These were all unfamiliar to the subject with which they were tested.

### Social facilitation control

In the social facilitation control of this study, the human remained in the enclosure; however, rather than placing the food dispenser outside the dog's enclosure, as in study 1, the food dis-penser was removed from the setup completely and turned off. If the subject pressed the but-ton in this condition, the human would not receive any food. This control condition differed from that in study 1, as it was deemed plausible that the novel position of the food dispenser, which was closer to the subject than to the human in study 1, might have altered subject's motivation by giving them the impression they had a better chance of obtaining the food com-pared with the test condition. The curtain also remained in place on the side of the subject's enclosure in the social facilitation control of study 2.

### Training

Two additional training stages were included at the end of the training phase of this study. The first new training stage matched the social facilitation control in terms of the setup (i.e. the

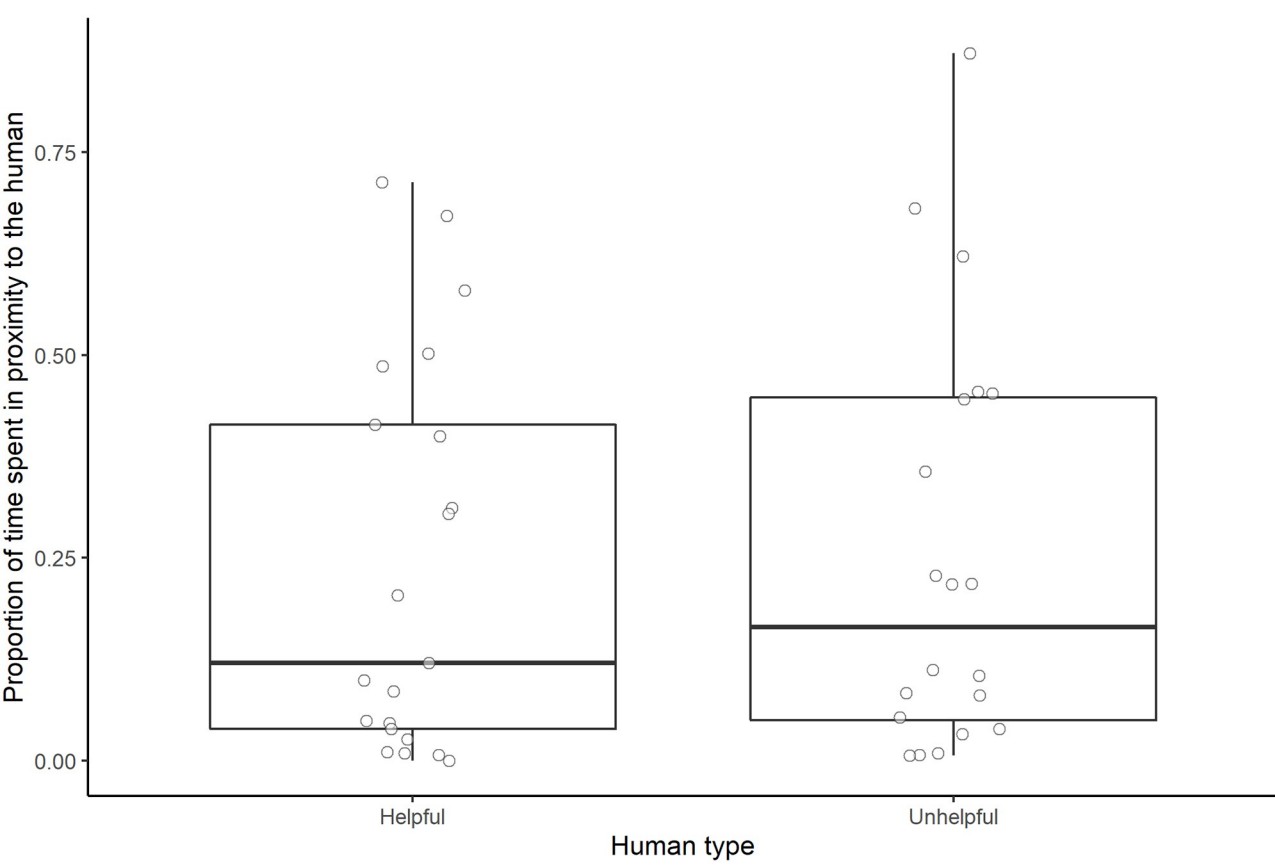

**Fig 8. Proportion of time spent in proximity to the two different human types.** Boxes display the interquartile range, black horizontal bars represent the median, whiskers represent the range of data points within 1.5 times the interquartile range from the upper and lower hinge, and circles represent individual data points.

subject had access to the button in its enclosure, and a human was present in the human enclosure but the food dispenser was not present). Ten trials were conducted with this setup. The subject was not required to press the button on any trial to proceed to the next stage. The next stage of training mimicked the asocial control condition in that the food dispenser was in the human's enclosure but no human was present. This also matched stage 5 of training. Again, ten trials were conducted and the subject was not required to press on any trial to proceed to the experiment. These training stages were both preceded and succeeded by a motivational session. The reason for inclusion of these new training stages was to ensure subjects had experienced each possible condition they would encounter in the test setting.

### Behaviour coding

Three experimenters coded the videos. All three experimenters also coded the same 23% of videos (approximately) for interobserver reliability coding.

### Statistical analyses

**Proportion of times the subjects pressed the button.** Analysis of the effect of human type, condition, and their interaction, on the proportion of times subjects pressed the button matched that for study 1 with a few differences. For instance, the fixed effect of "first human

**Table 2. Subject information including age, sex, breed, and the number of training sessions required to reach the criterion.**

| Subject ID | Age (years) | Sex | Breed | No. of sessions to reach criterion |
|---|---|---|---|---|
| A1 | 6 | F | Australian shepherd | 1 |
| B1 | 10 | F | Border collie | 1 |
| C1 | 7 | F | Border collie | 1 |
| D1 | 12 | F | Border collie | 1 |
| E1 | 8 | F | Border collie | 1 |
| F2 | 12 | M | Border collie | 1 |
| G2 | 5 | M | Border collie | 1 |
| H2 | 8 | F | Australian shepherd | 2 |
| I2 | 7 | F | Mix | 2 |
| J2 | 5 | M | Border collie mix | 1 |
| K2 | 7 | F | Golden retriever | 1 |
| L2 | 6 | M | Bernese mountain dog | 1 |
| M2 | 10 | M | Labradoodle | 2 |
| N2 | 4 | F | Mix | 2 |
| O2 | 5 | M | Border collie | 1 |
| P2 | 2 | M | Border collie | 1 |
| Q2 | 10 | M | German shepherd | 2 |
| R2 | 6 | F | Bernese mountain dog | 2 |
| S2 | 7 | F | Border collie mix | 1 |
| T2 | 3 | M | Petit Brabançon | 1 |
| U2 | 5 | M | Shetland sheepdog | 1 |

F, female; M, male.

experience" (i.e. whether the subject experienced the helpful or unhelpful human in their first experience phase) was excluded from this model and all other models in study 2. The reason for this exclusion was that the experience phase and test phase with a particular human took place on the same day, making first human experience redundant with the fixed effects of human type (helpful or unhelpful) and test day order (i.e. whether it was the first or second test day). In addition, a random slope of first human experience was not included within the random effect of human, and a random slope of condition was not included within the random effect of the dog-human dyad.

As with study 1, the full model here could only be fitted after exclusion of the correlations among random intercepts and random slopes. Also, for our full-null model comparison we chose the model which lacked the random slope of the interaction between condition and human type within the random effect of subject (log-likelihoods: model with complete set of random effects: -267.8015 [$df$ = 26]; model lacking observation level random effect: -267.8015 [$df$ = 25]; model lacking the random slope of the interaction between condition and human type within the random effect of subject: -267.8015 [$df$ = 24]; see S1 Appendix for the final full model).

The BLUPs did not deviate from a normal distribution, as confirmed by visual inspection of histograms [74, 78]. Collinearity did not appear to be an issue (maximum VIF: 1.016) and the model was not overdispersed (dispersion parameter = 0.494). The model was generally of acceptable stability with the possible exception of the interaction between condition and human type, the random intercept effects of subject, human and dog-human dyad, the random slopes of condition and human type within the random effect of human, and the random slope of human type within the random effect of subject (see S5 Fig).

The sample for this model included a total of 126 observations across 21 subjects, 24 humans, and 42 dyads. Interobserver reliability for the number of times the button was pressed was excellent (ICC = 0.994, $n_{observations}$ = 29, $n_{raters}$ = 3, $P < 0.001$).

**Latency to approach the human.**   The effect of human type on latency to approach the human was analysed in the same manner as study 1. We also selected the model with human as the frailty term, though the exact log-likelihood values differed (log-likelihoods: model with subject: -111.1651 [$df$ = 6.22]; model with human: -116.4762 [$df$ = 2.00]; see S1 Appendix for the final full model; for diagnostic plots see S6 and S7 Figs). The sample for this model included a total of 42 observations across 21 subjects and 24 humans. Interobserver reliability for the latency to approach the human was excellent (ICC = 0.988, $n_{subjects}$ = 10, $n_{raters}$ = 3, $P < 0.001$).

**Duration of proximity to the human.**   The effect of human type on the proportion of time subjects spent in proximity to the human in the free interaction session was analysed in the same manner as study 1 (see S1 Appendix for the final full model). The model was not overdispersed (dispersion parameter: 0.775) and collinearity did not appear to be an issue (maximum VIF: 1.002). Model stability was generally acceptable (see S8 Fig). The sample for this model included a total of 42 observations across 21 subjects and 24 humans. Interobserver reliability for the duration of proximity to the human was excellent (ICC = 0.983, $n_{subjects}$ = 10, $n_{raters}$ = 3, $P < 0.001$).

## Results

**Proportion of times the subjects pressed the button.**   Overall, neither condition, nor human type, nor the interaction between condition and human type had a significant effect on the proportion of times the subjects pressed the button (full-null model comparison: $\chi^2$ = 8.856, $df$ = 5, $P$ = 0.115; Fig 9).

**Latency to approach the human.**   There was no difference in the latency to approach the human in the interaction session based on whether they had been helpful or unhelpful (full-null model comparison: $\chi^2$ = 1.493, $df$ = 1, $P$ = 0.222; Fig 10).

**Duration of proximity to the human.**   Human type did not have a significant effect on the proportion of time subjects spent in proximity to the human in the interaction session (full-null model comparison: $\chi^2$ = 0.130, $df$ = 1, $P$ = 0.719; Fig 11).

## Discussion

Despite amending the procedure, dogs still did not reciprocate help received from humans. Although we reduced the complexity of the experience phase with regards to the activity of the humans and the number of buttons available, the new design may have introduced a confound: as the button was presented to the human in the same box as that used for subjects, it would have been difficult for subjects to see the button during the experience phase. Thus, it may not have been clear to the subjects that the human was pressing the button. Although this is plausible, the subjects could also see the movements of the human's arm, they were familiar with the sounds of the box opening and the sound emitted by the button when pressed, and dogs are capable of geometric perspective taking [98]. We, therefore, expect that understanding that the human was responsible for controlling the release of food from the dispenser was within the capabilities of the subjects.

It is, nonetheless, conceivable that subjects did not pay enough attention to the humans in the experience phase of either study to permit recognition of the action of the human. We investigated this possibility further by determining whether gazing at the human occurred at all. We focused on the experience phase of study 1 due to better video quality and, more

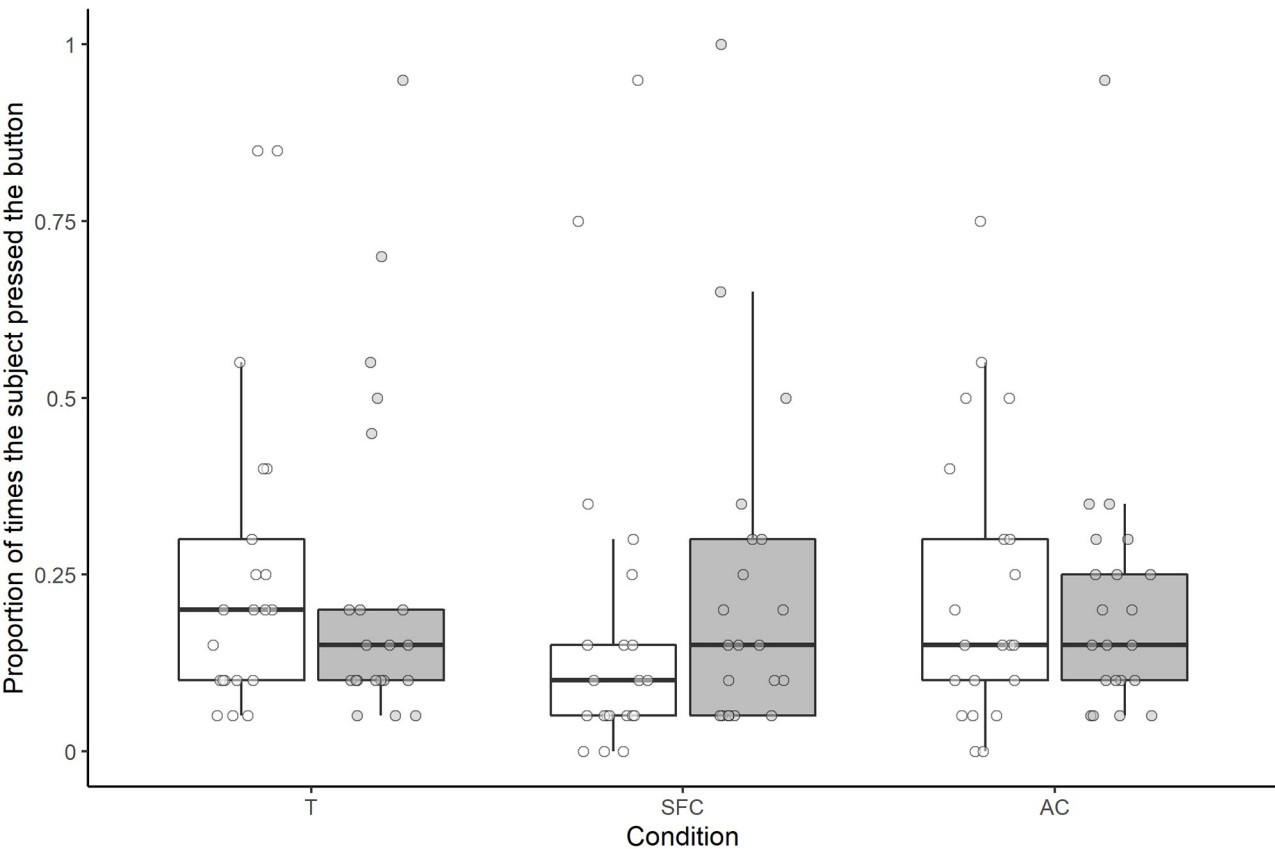

**Fig 9. Proportion of times the subjects pressed the button in each condition for each human type.** Pressing for the helpful human is represented in white and pressing for the unhelpful human is represented in grey. Boxes display the interquartile range, black horizontal bars represent the median, whiskers represent the range of data points within 1.5 times the interquartile range from the upper and lower hinge, and circles represent individual data points. T, test; SFC, social facilitation control; AC, asocial control.

specifically, the experience phase with the helpful human, as it is arguably the more important human to be attentive to in order to reciprocate.

Visual inspection of plotted data representing the proportion of time spent gazing at the helpful human immediately before button pressing (see S9 Fig) and immediately after button pressing (see S10 Fig) reveals that most subjects generally did look at or see the human in the experience phase for some duration, particularly before button pressing occurred. Much of this appeared to be due to the subjects already focusing on the button before the human pressed it, or due the subject tracking the human's hand when he/she began to move it towards the button. Gazing at the human's face did not seem common.

We also fitted a model to investigate whether the proportion of time spent gazing at the human changed across trials (see S1 Appendix for details). With increasing trial number, the proportion of time subjects spent looking at the helpful human before and after button pressing increased significantly. This finding may indicate that subjects were learning the importance of the human's action.

Regarding overall performance in the test phase, although we did not compare the results of the two studies statistically, median performance in terms of the number of times subjects pressed the button appeared lower in study 2 than in study 1 (see Figs 6 and 9). The reason for this difference is not clear, though it is possible that either the extra training steps in study 2, or

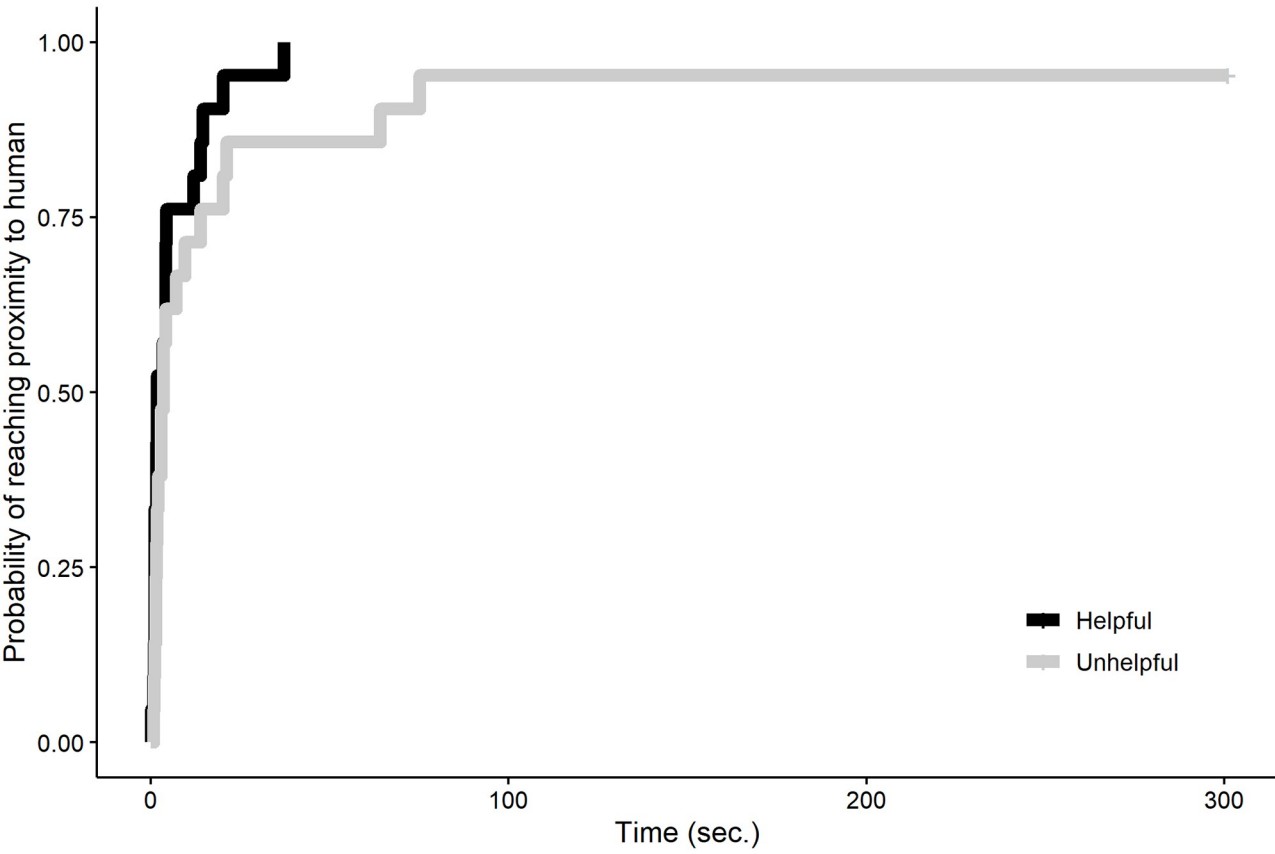

**Fig 10. Probability of reaching proximity to the human over time in the free interaction session.**

having an experience phase and a test phase on the same day, as was the case in study 2, reduced motivation to press in general in the test phase.

## General discussion

Across the two studies conducted here, dogs failed to reciprocate help received from the human. In addition, dogs did not appear to develop a preference for a particular human type (i.e. helpful or unhelpful) in the free interaction session, suggesting that either they did not distinguish between the two humans based on helpfulness or they did distinguish between them but developed no preference.

Our assessment of subjects' preference using the free interaction sessions was admittedly opportunistic and was also indirect, as subjects were never faced with a choice between the helpful and unhelpful human. As a result, our approach is arguably less sensitive than a direct preference test. However, the free interaction session offered the advantage, at least in study 2, of observing subjects interacting with the respective human at a standardized time point after the experience phase and test phase. Had a direct preference test been conducted, the experience with one human would have been more recent than the experience with the other, potentially biasing subjects' choice. Moreover, free interaction sessions like ours have been used to observe differential approach behaviour towards an experimenter based on experimental treatments immediately prior in inequity aversion studies [66, 99]. Thus, it seems likely that our

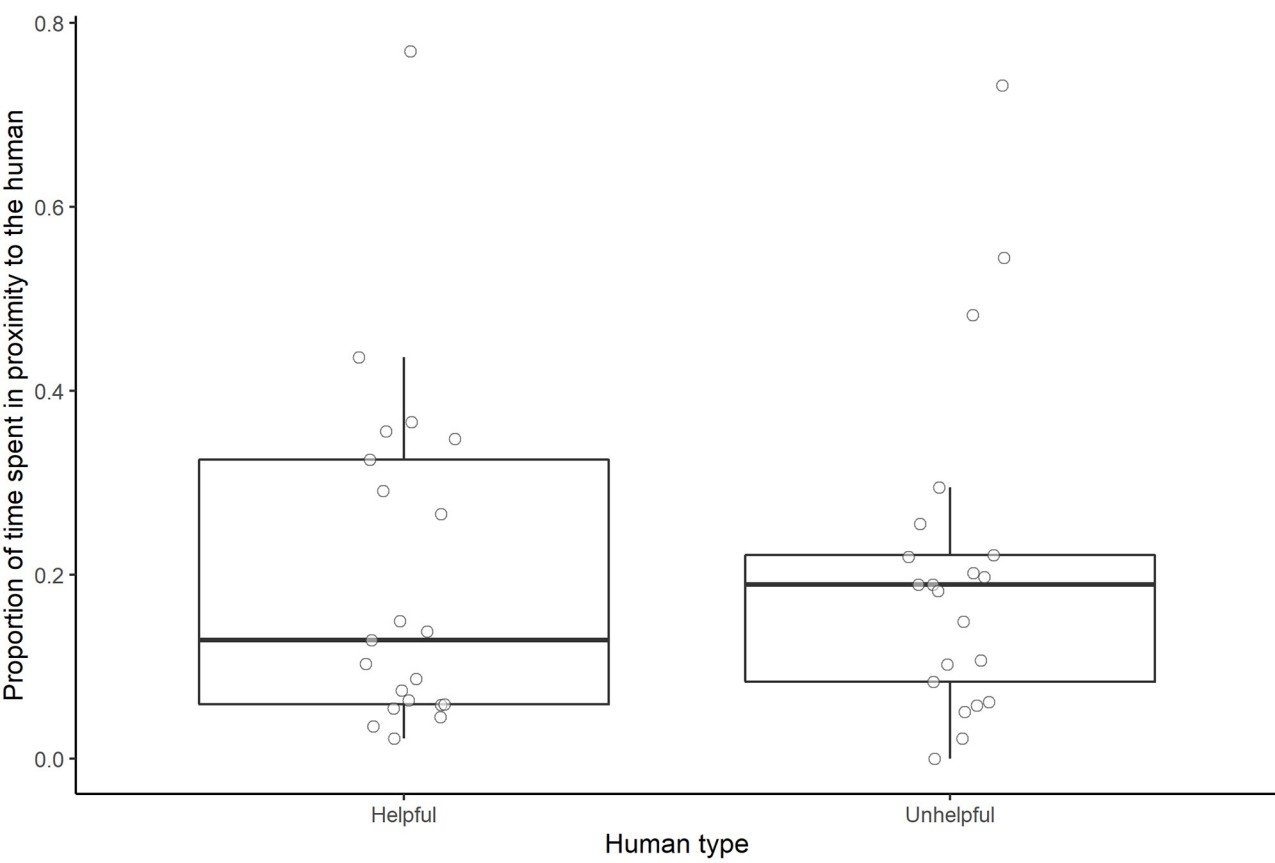

**Fig 11. Proportion of time spent in proximity to the two different human types.** Boxes display the interquartile range, black horizontal bars represent the median, whiskers represent the range of data points within 1.5 times the interquartile range from the upper and lower hinge, and circles represent individual data points.

assessment was sensitive enough to detect the subjects' preference (or lack thereof) for either human.

There are numerous potential explanations for the lack of reciprocity in the current studies, primarily related to the experimental approach. For example, the use of food as the currency here may have been inappropriate, as the provision of food in the dog-human relationship is typically asymmetrical, that is, humans usually provide pet dogs with food but dogs rarely provide humans with food. A similar explanation has been put forward (see Carballo et al. [23]) in relation to Quervel-Chaumette et al.'s [28] observation of dogs failing to act prosocially towards humans in a food-giving paradigm. However, the argument that food is inappropriate in such reciprocity studies between dogs and humans is not entirely satisfactory: adult dogs also do not typically provide each other with food; yet, in at least two experimental studies, they opted to provide conspecific partners with food [21, 22], and in an additional two studies, they even reciprocated the receipt of food from conspecifics [30, 35]. It is, notwithstanding, a worthwhile point that food as a resource in our studies may have been problematic; to the best of our knowledge no experimental study has shown prosociality from dogs to humans in a food-giving paradigm [28]. Two recent studies, in contrast, have demonstrated that dogs rescue their trapped and distressed owner from a box [23, 24]. Rescuing behaviour may, therefore, be a more appropriate choice for investigating reciprocal interactions between dogs and humans, though future studies will be required to determine whether such behaviour actually

represents a prosocial act of rescuing or an attempt to seek proximity to the owner in a time of distress.

The lack of a preference for either human in the free interaction session, in both studies, suggests that the lack of reciprocity may have little to do with dogs' propensity to provide humans with food and more to do with a failure to recognize the helpful act of the human. A variety of interrelated explanations might account for dogs' failure to recognize the cooperative act of the human. First, the method of food provision in our studies involved an effectively abstract connection between a button and a food dispenser. Most studies demonstrating successful food provisioning in dogs so far have used methods involving clear physical relationships (e.g. a rope connected to a tray [21, 30, 35]; a lever which can be pressed to open a box [35]). The abstract connection between the button and the dispenser may have been a problem here.

Two experimental studies have demonstrated prosociality in dogs using abstract connections similar to ours. First, Bräuer et al. [25] demonstrated that dogs can press a button which leads to the opening of a door allowing a human to gain access to a compartment. However, prosocial pressing of the button was not spontaneous, as it only occurred in response to pointing and natural communication from a human. Although these results suggest that the dogs were not prosocial of their own volition, the extent to which the abstract connection between the pressing of the button and the outcome influenced the findings is unclear. Second, Dale et al. [22] applied a token choice task to assess dogs' prosocial tendencies towards both familiar and unfamiliar conspecifics. Unlike Quervel-Chaumette et al.'s [21] study in which a rope connected to a tray could be pulled to deliver food to the conspecific partner, subjects could choose between two tokens, one of which resulted in the partner being rewarded with food. Dale et al. [22] obtained very similar results to Quervel-Chaumette et al. [21] in that the dogs opted to provide the familiar conspecific, but not the unfamiliar conspecific, with food. This result would seem to suggest that abstract connections should not interfere with dogs' prosocial choices in experimental tasks; however, in contrast with the study of Quervel-Chaumette et al. [21], subjects also opted for the prosocial choice in the social facilitation control, suggesting they may not have understood the task fully. Any such lack of understanding of the task is not necessarily linked to the abstract connection between the required behaviour and the food delivery mechanism, but it does at least raise suspicions that an abstract connection could be an issue.

Two recent prosociality studies characterized by similarly abstract connections between the required behaviour and the food delivery (one based on a touch screen task [100] and the other based on choosing a specific location [101]) failed to demonstrate prosocial preferences towards conspecifics in dogs (though wolves, the closest living relatives of dogs [102], were prosocial in the touch screen task [100]). It is similarly unclear from these studies whether the abstractness of the task explains the absence of prosociality in dogs. Nevertheless, even if abstract relationships do not hinder prosociality in dogs, dogs may still have trouble perceiving or understanding the helpful act of *another* individual that is using such an abstract mechanism.

An additional possible explanation for subjects' potential lack of registering the cooperative act of the human, is that the behaviour performed by the human was not a natural cooperative behaviour. There are two clear aspects common to natural cooperative interactions which were missing here. First, typical behaviours that are performed prosocially and reciprocated in nature include those such as grooming and active food sharing [1]. For the most part, these involve clear physical interactions between individuals; thus, the behaviour of the helpful individual is likely to be quite salient. In our studies, the human was always separated from the subject by a fence and performed a behaviour that did not require any physical contact with

the subject. Moreover, the food itself was delivered by, and eaten from, the food dispenser. Thus, due to the lack of physical contact between the human and the subject, the salience of the human and their cooperative or non-cooperative act may have been quite low. Furthermore, cooperative behaviours that do not necessarily involve physical interactions, such as group defence, if natural social behaviours for the species in question, are likely to be highly salient. Nevertheless, the behaviour performed here by the human did not resemble any cooperative behaviour within the natural repertoire of dogs.

Second, natural interactions typically involve communicative signals between interacting individuals. In our studies, the human refrained from active communication with the subject throughout the experience phase. The absence of natural communication was also highlighted by Quervel-Chaumette et al. [28] as a potential explanation for dogs' failure to behave prosocially towards humans in their food-giving setup. Furthermore, dogs were only prosocial towards humans in the study of Bräuer et al. [25] when natural communication or pointing cues were introduced. It is plausible that more natural communication delivered by the human to the dog in our studies would have increased the salience of the human and their activity. However, we specifically chose not to include natural communication to avoid the possible coercive influence of human communication and the potential introduction of other unintentional cues.

Although both of the above arguments (i.e. the absence of a natural cooperative behaviour and physical contact, and the absence of natural communication) seem quite plausible, particularly in combination, it would be surprising to some extent if these accounted for our findings in whole. In both studies of Gfrerer & Taborsky [30, 35], in which dogs reciprocated the receipt of food from conspecific partners, the food was initially delivered using an artificial mechanism, and subsequently reciprocated in a similar manner. Thus, like our studies, no physical contact occurred between the subject and partner, and natural cooperative behaviours were not employed. Nevertheless, future studies investigating reciprocal interactions between dogs and humans may benefit from using natural behaviours involving physical contact such as active food sharing or grooming during the experience phase and allowing natural communication between the human and the dog.

Finally, it is possible that, as a result of the training, the button and the dispenser were more salient than the human and occupied most of the dogs' attention throughout both studies. Consequently, the dogs may not have paid enough attention to the human to register fully the relevance of the human's actions, even though most subjects did at least look at or see the human to some degree. Increasing the salience of the human and the human's actions could overcome this issue and facilitate the subjects' registering of the cooperative act.

In this context, it is conceivable that the limited training and exposure to the action of the humans also diminished the dogs' likelihood of perceiving or understanding the human's action. Gfrerer and Taborsky's [30, 35] training procedure, which included providing a partner with food and receiving food from a partner, required approximately 14 to 19 days with two sessions per day for each subject. Moreover, they incorporated the exchange of roles with a partner. The training protocol in experimental studies in which rats were observed to reciprocate was similar, with repeated experience of exchanging roles prior to the experimental procedure [103, 104]. The drawn out experience of the receipt of food from a partner, and providing food to a partner, over a number of days, combined with the rapid exchange of roles, may have facilitated an understanding of the significance of the partner's cooperative or non-cooperative behaviour. In contrast, in our design, the subjects had a single experience day with each human, albeit consisting of numerous trials. Extensive training may be required to facilitate subjects' attentiveness to the crucial aspects of the setup.

In the context of learning about crucial aspects of the setup, it is worth noting that to reciprocate directly in our setup, dogs would have needed to discriminate between the helpful and unhelpful human. The discrimination learning required may have been too demanding for such a small number of sessions. Visual and olfactory discrimination learning in dogs can take a considerable number of training sessions [97, 105, 106]. In fact, in a discrimination learning task in which dogs had to discriminate between their owner and a familiar person based on visual information from the faces and heads, an average of 6.5 sessions, each lasting ten trials, were required for dogs to successfully discriminate [107]. Similarly, dogs were found to require between three and eight sessions of 20 trials each to successfully discriminate between their owner's and a stranger's face [108]. Moreover, discrimination between unfamiliar humans in our studies may have been particularly demanding. Increasing the perceptual differences between the humans may help with learning the discrimination in future studies.

Despite the apparent difficulty of discrimination learning, however, it is worth keeping in mind that dogs were previously shown by Gfrerer and Taborsky [35] to reciprocate after just 14 trials of experience with each partner type. It is plausible that the more extensive training procedure, outlined above, facilitated discrimination by priming subjects to attend to the partner. Furthermore, partner discrimination may be easier if the partners are conspecifics. However, as subjects received an experience phase and test phase with a single partner in a two-day sequence before encountering a new partner [35], discrimination learning may not have been necessary to successfully reciprocate. Nonetheless, Heberlein et al. [56] and Carballo et al. [57] found that dogs could discriminate between humans that differed in cooperativeness after just six and twelve trials respectively. As subjects in our studies were presented with each human across at least 20 trials, it is not clear that the challenge of discrimination learning can account for the lack of reciprocity.

The explanations presented here for the failure of subjects to reciprocate tend to imply that dogs would have reciprocated were it not for methodological issues. However, it is possible that dogs do not have a predisposition to engage in reciprocal cooperation with humans naturally. In fact, a recent study comparing inequity aversion in dog breed groups that vary in the extent to which they were selected for cooperation with humans, suggested that dog-human cooperation may not involve this form of reciprocity [109]. It is also worth keeping in mind that most instances of dog-human cooperation involve extensive training combined with the exploitation of already selectively bred behaviours. Thus, the extent to which standard explanations for the evolution of cooperation apply to cooperative interactions between dogs and humans is undoubtedly limited. Moreover, our results corroborate findings from other studies in which dogs failed to behave prosocially towards humans [25–27, 29], especially in a food-giving task [28]. It is, nonetheless, unclear why dogs would not extend their experimentally demonstrated reciprocal capacities to interactions with humans. One way to disentangle the species-specific nature of our results from the potential methodological issues could be to repeat our experiments using dogs as the helpful and unhelpful partners rather than humans.

In future studies, the familiarity of the partners may also need to be considered. Our use of unfamiliar partners prevented results from being confounded by previous experience with those specific individuals. However, familiarity appears to influence prosociality in dogs, at least in experimental tasks with conspecifics [21, 22]. Thus, the use of unfamiliar partners may have precluded any prosocial food-giving by the dogs. In addition, much reciprocity in non-human animals is thought to occur within long-term affiliative relationships [53]. It may, therefore, be advantageous to study reciprocity between dogs and humans by using humans with whom the dogs are already familiar.

In conclusion, dogs in the current studies failed to reciprocate help received from humans and a preference for either human type (helpful or unhelpful) was not evident based on a free

interaction session. Given that dogs have already been shown to reciprocate help received from conspecifics in experimental studies, the absence of reciprocity here may be explained by methodological inadequacies, though it is also possible that dogs are not predisposed to engage in such cooperative interactions with humans naturally. Future studies investigating direct reciprocity between dogs and humans may benefit from the application of rescuing as the cooperative behaviour required of the dog, and a natural behaviour such as food-giving or grooming, involving physical contact and active communication, as the cooperative behaviour provided by the human. In addition, a more extensive training procedure than that applied here, including the repeated exchange of roles between the dog and the human may facilitate the emergence of reciprocity. Moreover, increasing the difference between the helpful and unhelpful partners could aid the subjects in discriminating between them. Finally, repetition of the current setup but with intraspecific rather than interspecific interactions would help determine whether the current results reflect dogs' disinclination to reciprocate help received from humans, or whether they are best explained by methodological issues.

## Supporting information

**S1 Fig. Model stability plot.** Model stability plot for the proportion of times subjects pressed the button, representing the range of model estimates for each term in the model when the levels of the random effects are excluded one at a time. Names including an ampersat symbol (@) refer to random effects: the first term in the name is the grouping variable; the term after the first ampersat is either a random intercept (indicated by "(Intercept)") or a random slope. (SVG)

**S2 Fig. Scaled Schoenfeld residuals.** Scaled Schoenfeld residuals [110, 111] plotted against time to event, for each fixed effect, to assess the assumption of proportional hazards in the Cox proportional hazards model assessing latency to reach proximity to the human. The solid line is a smoothing line fit to the plot and the dashed lines create a +/- 2 standard error band. (SVG)

**S3 Fig. Dfbeta residuals.** Dfbeta values [112] for each observation for each fixed effect in the Cox proportional hazards model assessing latency to reach proximity to the human. The red dashed line is a horizontal line representing a dfbeta value of 0 (see Kassambara et al. [85]). (SVG)

**S4 Fig. Model stability plot.** Model stability plot for duration of proximity to the human, representing the range of model estimates for each term in the model when the levels of the random effects are excluded one at a time. Names beginning with "cond" refer to the mean part of the model. The name beginning with "disp" refers to the dispersion part of the model. Names including more than one ampersat symbol (@) refer to random intercepts. (SVG)

**S5 Fig. Model stability plot.** Model stability plot for the proportion of times subjects pressed the button, representing the range of model estimates for each term in the model when the levels of the random effects are excluded one at a time. Names including an ampersat symbol (@) refer to random effects: the first term in the name is the grouping variable; the term after the first ampersat is either a random intercept (indicated by "(Intercept)") or a random slope. (SVG)

**S6 Fig. Scaled Schoenfeld residuals.** Scaled Schoenfeld residuals [110, 111] plotted against time to event, for each fixed effect, to assess the assumption of proportional hazards in the Cox proportional hazards model assessing latency to reach proximity to the human. The solid line

is a smoothing line fit to the plot and the dashed lines create a +/- 2 standard error band.
(SVG)

**S7 Fig. Dfbeta residuals.** Dfbeta values [112] for each observation for each fixed effect in the
Cox proportional hazards model assessing latency to reach proximity to the human. The red
dashed line is a horizontal line representing a dfbeta value of 0 (see Kassambara et al. [85]).
(SVG)

**S8 Fig. Model stability plot.** Model stability plot for duration of proximity to the human, representing the range of model estimates for each term in the model when the levels of the random effects are excluded one at a time. Names beginning with "cond" refer to the mean part of
the model. The name beginning with "disp" refers to the dispersion part of the model. Names
including more than one ampersat symbol (@) refer to random intercepts.
(SVG)

**S9 Fig. Proportion of time spent gazing at the human before button pressing, across trials.**
Transparent dots represent data points. The solid line shows the fitted model and the dotted
lines show its 95% confidence interval.
(SVG)

**S10 Fig. Proportion of time spent gazing at the human after button pressing, across trials.**
Transparent dots represent data points. The solid line shows the fitted model and the dotted
lines show its 95% confidence interval.
(SVG)

**S1 Appendix.**
(DOC)

**S1 Dataset.**
(XLSX)

**S1 Video.**
(MP4)

# Acknowledgments

We would like to thank the dogs and their owners for participation in this study. We are also
very grateful to all those who participated in the study as helpful and unhelpful human partners. We thank Roger Mundry for statistical support. We also thank Karin Bayer, Jennifer
Bentlage, Karoline Bürger, Aleksandar Orlic, Brigitte Pavlik, and Christina Szaga-Doktor for
administrative support, as well as Peter Füreder and Wolfgang Berger for technical support.

# Author Contributions

**Conceptualization:** Jim McGetrick, Friederike Range.

**Data curation:** Jim McGetrick, Lisa Poncet, Marietta Amann, Johannes Schullern-Schrattenhofen, Leona Fux, Friederike Range.

**Formal analysis:** Jim McGetrick.

**Funding acquisition:** Jim McGetrick, Friederike Range.

**Investigation:** Lisa Poncet, Marietta Amann, Johannes Schullern-Schrattenhofen, Leona Fux,
Mayte Martínez.

**Methodology:** Jim McGetrick, Lisa Poncet, Marietta Amann, Johannes Schullern-Schrattenhofen, Leona Fux, Mayte Martínez.

**Project administration:** Jim McGetrick, Mayte Martínez, Friederike Range.

**Resources:** Jim McGetrick, Friederike Range.

**Supervision:** Jim McGetrick, Mayte Martínez, Friederike Range.

**Validation:** Jim McGetrick.

**Visualization:** Jim McGetrick.

**Writing – original draft:** Jim McGetrick.

**Writing – review & editing:** Jim McGetrick, Lisa Poncet, Marietta Amann, Johannes Schullern-Schrattenhofen, Leona Fux, Mayte Martínez, Friederike Range.

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
