## [Decision Letter · Decision Letter 0]

11 Dec 2020

PONE-D-20-35864

Dogs fail to reciprocate the receipt of food from a human in a food-giving task

PLOS ONE

Dear Dr. McGetrick,

Thank you for submitting your manuscript to PLOS ONE. After careful consideration, we feel that it has merit but does not fully meet PLOS ONE’s publication criteria as it currently stands. Therefore, we invite you to submit a revised version of the manuscript that addresses the points raised during the review process.

It was reviewed by two experts in the field and they have recommended a few modifications be made prior to acceptance.

I therefore invite you to consider the suggested changes and resubmit your manuscript, along with a response to reviewers to aid re-revision.

We look forward to receiving your revised manuscript.

I wish you the best of luck with your revisions.

Hope you are keeping safe and well in these difficult times.

Kind regards,

Simon Clegg, PhD

Academic Editor

PLOS ONE

Reviewers' comments:

Reviewer's Responses to Questions

**Comments to the Author**

1. Is the manuscript technically sound, and do the data support the conclusions?

Reviewer #1: Yes

Reviewer #2: Yes

2. Has the statistical analysis been performed appropriately and rigorously? 

Reviewer #1: Yes

Reviewer #2: Yes

3. Have the authors made all data underlying the findings in their manuscript fully available?

Reviewer #1: Yes

Reviewer #2: Yes

4. Is the manuscript presented in an intelligible fashion and written in standard English?

Reviewer #1: Yes

Reviewer #2: Yes

5. Review Comments to the Author

Reviewer #1: This is an interesting, relevant and original work. The manuscript is clear and well-written and the authors have tried to do a thorough analysis of the results. They found that dogs were not able to reciprocate the help received from humans. Even when the lack of this ability in a food-giving task is a possibility that this work as well as other previous studies have suggested, the limitations in the experimental designs force us to be cautious.

The authors clearly described and analyzed some of these limitations. However, another plausible and parsimonious explanation of the results is that dogs were repeatedly reinforced throughout the procedure for pressing the button. So, they could continue pressing the button, just because this behavior had a strong association with the food, regardless of the rest of the conditions. In addition, considering that reinforced and not reinforced trials were mixed (because of the motivational sessions) in the design, dogs were exposed to a partial reinforcement schedule, which increases the resistance to extinction. In this way, the association between pressing the button and food could be higher than anything else. Moreover, in order to successfully solve the task, the dogs must perform a discrimination learning in which they have to inhibit their learned response of pressing the button in some contexts. This kind of learning usually requires long trainings. You have to state this limitation in the discussion.

Anyway, I think this problem always appears when you have to train the behavior required to help the other individual. For this reason, it is better to choose a response that is already part of the repertoire of the animals.

On the other hand, one crucial aspect in this kind of experiments is whether dogs are able to understand the situation. In this sense, an overshadowing phenomenon may have occurred. As a consequence of the training, the device (button and dispenser) was very salient. So, when dogs pressed the button, it is likely that they were paying more attention to the device than to the humans. If they paid little or no attention to them, the likelihood of learning about their behavior is low. Do you have any assessment of the behavior of the dogs during the conditions in which humans were present? Can you assess gaze duration towards humans?

As far as I understand, you did not perform a preference test between the helpful and the unhelpful human, so you cannot conclude that dogs were not able to discriminate them. Your comparison is indirect and therefore less sensitive. The length of time in which dogs were near each person compared to the owner is not a good indicator of the helper-unhelper discrimination. For example, the presence of the owner during the free interaction session could interfere, decreasing the negative reaction towards the unhelpful human, decreasing the interest in the unfamiliar person, etc. You have to relativize your conclusions in this regard.

It is necessary to include some important information about the characteristics of the sample: the description of the breeds and the level of previous training.

Which was the reward used? I imagine that the helpful human gave the dog a better reward than the one the dog obtained by itself during training and motivational trials.

I can see by comparing Fig 6 and 9 that the performance of the dogs decreased between the two studies. Can you explain this difference?

I think that the legend of Fig 1 is too long and includes many details that are already included in the text. I suggest reducing it.

Please describe the interval between phases.

L 324-325 I do not understand the objective of modifying the first trial procedure.

L416 “The video for one of the free interaction sessions was unavailable” Do you mean the video from one dog or the videos of all the dogs during one session? In this case, which was the session?

426 and 551 vs 427. Did you use the number of times or the proportion of times?

Reviewer #2: General comments

In the present study authors tested if dogs would reciprocate a pro-social act from an unknown human in a context involving food, using a novel experimental design adapted from previous studies in dogs and other species. In the first study the authors did not find any evidence of reciprocity in dogs and speculate that this result could have been due to the complexity of the task or the mnemonic demands imposed to the dogs by the time gap between the training and testing phases. Thus, in a second study the authors simplify the task and run the training and testing phases in the same day. Again, dogs failed to reciprocate the human pro social behavior in this simplified version of the task. These findings led the authors to conclude that dogs do not spontaneously reciprocate the human pro-social behavior in contexts involving food.

I want to congratulate the authors for their work; I found the paper very interesting and carefully design. I will suggest some minor revisions that could probably improve the paper.

Specific comments

Introduction

The review of the previous literature is precise updated and well written. Nevertheless:

L. 43-44. I would remove the phrase. “Reliable models of reciprocity that can be

manipulated experimentally are scarce”. There is a vast literature on reciprocity on rats, mice, and insects. And I believe that there are all valuable models for the study of reciprocity (e.g. Taborsky’s works).

L. 89 -102. I found this paragraph a bit unnecessary. Although, as mention, all the listed characteristics made the dog a good model for the study of pro-sociality in general these are not specific characteristics needed for reciprocity. I would focus in reciprocity (See. Trivers, 1971).

L. 160-161. In line with the previous comment. Some of the aspects that would enhance direct reciprocity are: the possibility of future encounters (Trivers, 1971), and the bond between the subjects (i.e. De Wal, 1997; Freidin et al., 2013). So I think that one big difference with some previous studies (Quervel-Chaumette et al., 2015; 2016, Sanford et al., 2018; Carballo et al., 2020) is the identity of the receiver. Most evidences showing spontaneous pro social behavior in dogs (or the lack of it, Quervel-Chaumette 2016) came from studies with the owners, or some familiar human, as partner. I understand the methodological choice of the authors to use unknown humans as partners which eliminate a lot of uncontrolled variations that could mixed the results. But I believe that the best solution would be to run a third study whit the owners as partners and counterbalance their role (Helper vs. Unhelper) between subjects. The inclusion or non-inclusion of this third study would not influence in my personal decision to suggest the publication of the paper. I simply mention that this would be an interesting aspect to take into account and probably the authors would like to include a further study or mention this aspect in the conclusion.

Methods and analysis

I found the methodology very well design and implemented. I think the data presented by the authors in the data set of supplementary material and the variables analyzed are clear and with a straightforward interpretation. So I found the statistical analyses unnecessarily a bit too complex for the data and the number of factors taken in consideration. But authors may well prefer to keep the presented analyses.

L. 261. Who was the human who eat the food?

Discussion

Again, I found the discussion well written and pertinent. And I mostly agree with the authors’ interpretation of the data. Nevertheless, I would include some references to the role of learning mechanisms in the development of pro social behavior in dogs and in reciprocity in particular. The authors mentions this when they speculate that with further training and switching roles between dogs and humans dogs probably could reciprocate the human but they do not provide and explicit theoretical link. For example, dogs could be focusing in some other aspect of the experimental situation which could overshadow the role of the helper, thus making the actions of the helper more salient could improve dogs performance. Furthermore, the discriminations between unknown humans could be demanding for dogs, thus increasing the difference between helper and un-helper could also improve dogs’ performance or the testing could be done with familiar humans. These aspects are related to the factors that could facilitate reciprocity that I mention earlier (familiarity, bonding, repetition of the encounters etc.) and are usually taken into account when discussing pro-social behavior in dogs.

L. 885: include “direct” before reciprocity.

6. PLOS authors have the option to publish the peer review history of their article (what does this mean?). If published, this will include your full peer review and any attached files.

Reviewer #1: No

---

## [Author Response · Author response to Decision Letter 0]

17 May 2021

Reviewer #1: This is an interesting, relevant and original work. The manuscript is clear and well-written and the authors have tried to do a thorough analysis of the results. They found that dogs were not able to reciprocate the help received from humans. Even when the lack of this ability in a food-giving task is a possibility that this work as well as other previous studies have suggested, the limitations in the experimental designs force us to be cautious.

The authors clearly described and analyzed some of these limitations. However, another plausible and parsimonious explanation of the results is that dogs were repeatedly reinforced throughout the procedure for pressing the button. So, they could continue pressing the button, just because this behavior had a strong association with the food, regardless of the rest of the conditions. In addition, considering that reinforced and not reinforced trials were mixed (because of the motivational sessions) in the design, dogs were exposed to a partial reinforcement schedule, which increases the resistance to extinction. In this way, the association between pressing the button and food could be higher than anything else. Moreover, in order to successfully solve the task, the dogs must perform a discrimination learning in which they have to inhibit their learned response of pressing the button in some contexts. This kind of learning usually requires long trainings. You have to state this limitation in the discussion. 

We have now included two paragraphs in the discussion on discrimination learning. The first points out that discrimination learning would have been required for successful reciprocation here and that discrimination learning requires many sessions in studies with dogs. The second paragraph points out that although discrimination learning clearly takes a long time in dogs based on discrimination learning studies, dogs were able to discriminate between a cooperative and uncooperative partner after a very small number of trials in some studies:

L899 – 922: In the context of learning about crucial aspects of the setup, it is worth noting that to reciprocate directly in our setup, dogs would have needed to discriminate between the helpful and unhelpful human. The discrimination learning required may have been too demanding for such a small number of sessions. Visual and olfactory discrimination learning in dogs can take a considerable number of training sessions (97,105,106). In fact, in a discrimination learning task in which dogs had to discriminate between their owner and a familiar person based on visual information from the faces and heads, an average of 6.5 sessions, each lasting ten trials, were required for dogs to successfully discriminate (107). Similarly, dogs were found to require between three and eight sessions of 20 trials each to successfully discriminate between their owner’s and a stranger’s face (108). Moreover, discrimination between unfamiliar humans in our studies may have been particularly demanding. Increasing the perceptual differences between the humans may help with learning the discrimination in future studies. 

Despite the apparent difficulty of discrimination learning, however, it is worth keeping in mind that dogs were previously shown by Gfrerer and Taborsky (35) to reciprocate after just 14 trials of experience with each partner type. It is plausible that the more extensive training procedure, outlined above, facilitated discrimination by priming subjects to attend to the partner. Furthermore, partner discrimination may be easier if the partners are conspecifics. However, as subjects received an experience phase and test phase with a single partner in a two-day sequence before encountering a new partner (35), discrimination learning may not have been necessary to facilitate reciprocity. Nonetheless, Heberlein et al. (56) and Carballo et al. (57) found that dogs could discriminate between humans that differed in cooperativeness after just six and twelve trials respectively. As subjects in our studies were presented with each human across at least 20 trials, it is not clear that the challenge of discrimination learning can account for the lack of reciprocity.

It is plausible that the training procedure caused the button and dispenser to become very salient for the dogs, thereby reducing their attentiveness to the human. We have included discussion points on this (see below). However, we did not get ceiling effects with regards to the number of times the dogs pressed the button. In fact, in study 1 and study 2 performance was relatively low e.g. median number of presses of ~7 – 10 in study 1 and ~2 – 4 in study 2, indicating that subjects were well able to inhibit pressing. Furthermore, at least 2 other studies demonstrating prosociality in dogs in food-giving tasks used similar training procedures to ours (see Quervel-Chaumette et al. 2015 and Dale et al. 2016) and the training did not appear to inhibit the subjects’ ability to make the decision to provide the partner with food. Thus, we do not believe the reinforcement schedule had a major impact on the outcome of either study.

Anyway, I think this problem always appears when you have to train the behavior required to help the other individual. For this reason, it is better to choose a response that is already part of the repertoire of the animals. On the other hand, one crucial aspect in this kind of experiments is whether dogs are able to understand the situation. In this sense, an overshadowing phenomenon may have occurred. As a consequence of the training, the device (button and dispenser) was very salient. So, when dogs pressed the button, it is likely that they were paying more attention to the device than to the humans. If they paid little or no attention to them, the likelihood of learning about their behavior is low. Do you have any assessment of the behavior of the dogs during the conditions in which humans were present? Can you assess gaze duration towards humans? 

We have now coded the duration of gaze at the helpful human in the experience phase of study 1 for the periods immediately before button pressing and immediately after button pressing. We fitted a model to analyse the effect of trial number on gazing. The following details covering the video coding and analysis are included in S1 Appendix:

Gazing at the helpful human in the experience phase

Methods

Coding

The duration of gaze at the helpful human in experience phase sessions was coded. Instances of looking at or seeing any part of the human’s body (e.g. seeing the human’s hand on the button) were coded. Coding for gaze duration was divided among five experimenters and for interobserver reliability, all five experimenters coded the same 20% of the videos. Six trials from one session with one subject were missing due to the absence of a video recording.

Statistical analysis

For each trial we determined the proportion of time spent gazing at the helpful human in the 3 second period immediately before button pressing and in the 3 second period immediately after button pressing. For one single trial with one subject, there were less than 3 seconds on video prior to button pressing; therefore, the total duration used to determine the proportion was adjusted.

To determine whether trial number had an influence on the proportion of time subjects spent looking at the helpful human before button pressing, we fitted a Generalized Linear Mixed Model (GLMM) with a beta error distribution and a logit link function (1–3). As such beta models cannot handle zeroes and ones, and our dataset comprised both, we transformed the response using the following formula (where “x” is the variable to be transformed; see Smithson and Verkuilen, 3): 

x’ = x.(length(x)-1) + 0.5

 length(x)

We included the fixed effects of trial and trial squared. We included random intercept effects of subject and human, and random slopes of trial and trial squared were included within both.

We fitted the model using the function “glmmTMB” from the package “glmmTMB” (version 1.0.2.1; 4). Prior to fitting the model, and prior to squaring trial number, we z-transformed trial number to a mean of zero and a standard deviation of one to allow for an easier interpretation of results and to ease model convergence. All correlations between random slopes and random intercepts were removed due to convergence issues. The model was not overdispersed (dispersion parameter: 0.6989). A full-null model comparison was carried out, as an overall test of the effect of trial number and trial number squared on the proportion of time spent looking at the helpful human before button pressing. 

Confidence intervals of model coefficients were derived using 1,000 parametric bootstraps using the function “simulate” of the package “glmmTMB” (version 1.0.2.1; 4) and a wrapper kindly provided by Roger Mundry. Tests of the individual fixed effects were derived with likelihood ratio tests (5) by using the R function “drop1” and setting the argument “test” to “Chisq”.

The effect of trial on the duration of gaze at the human after button pressing was analysed using an identical model structure; however, no random slopes were included due to convergence issues. This model was not overdispersed (dispersion parameter: 0.8290). The sample for both models included a total of 1,044 observations across 21 subjects and 12 humans. Interobserver reliability for duration of gaze at the helpful human before and after button pressing was moderate (before: ICC = 0.725, nobservations = 193, nraters = 5, P < 0.001; after: ICC = 0.61, nobservations = 193, nraters = 5, P < 0.001). Models were plotted using R (version 4.0.2; 6).

Results

Proportion of time spent gazing at the human before button pressing

Overall, there was a significant effect of the fixed effects on the proportion of time spent gazing at the helpful human before button pressing (full-null model comparison: χ2 = 11.437, df = 2, P = 0.003). More specifically, with a greater number of trials, the proportion of time spent gazing at the helpful human increased (see S1 Table and S1 Fig).

S1 Table Results of the model analysing the effects of trial and trial squared on the proportion of time subjects spent gazing at the helpful human before button pressing in the experience phase. Estimates are presented along with standard errors, confidence intervals, and likelihood ratio test results.

Term Estimate SE Lower CI Upper CI χ2 df P

Intercept -0.794 0.214 -1.209 -0.391 - - -

Triala 0.290 0.084 0.117 0.460 8.792 1 0.003

Trial squareda -0.084 0.046 -0.174 0.010 2.668 1 0.102

aTrial was z-transformed to a mean of 0 and a standard deviation of 1 prior to inclusion in the model and prior to squaring.

Proportion of time spent gazing at the human after button pressing

Overall, a trend was revealed for the influence of the fixed effects on the proportion of time spent gazing at the helpful human after button pressing (full-null model comparison: χ2 = 5.569, df = 2, P = 0.062). There was a weak effect whereby the proportion of time spent gazing at the human increase with more trials (see S2 Table and S2 Fig).

S2 Table Results of the model analysing the effects of trial and trial squared on the proportion of time subjects spent gazing at the helpful human before button pressing in the experience phase. Estimates are presented along with standard errors, confidence intervals, and likelihood ratio test results.

Term Estimate SE Lower CI Upper CI χ2 df P

Intercept -2.310 0.126 -2.572 -2.083 - - -

Triala 0.065 0.032 0.002 0.130 4.019 1 0.045

Trial squareda 0.043 0.036 -0.026 0.111 1.408 1 0.235

aTrial was z-transformed to a mean of 0 and a standard deviation of 1 prior to inclusion in the model and prior to squaring.

We have discussed this in the discussion for study 2:

L744 – 761: It is, nonetheless, conceivable that subjects did not pay enough attention to the humans in the experience phase of either study to permit recognition of the action of the human. We investigated this possibility further by determining whether gazing at the human occurred at all. We focused on the experience phase of study 1 due to better video quality and, more specifically, the experience phase with the helpful human, as it is arguably the more important human to be attentive to in order to reciprocate. 

Visual inspection of plotted data representing the proportion of time spent gazing at the helpful human immediately before button pressing (see S9 Fig) and immediately after button pressing (see S10 Fig) reveals that most subjects generally did look at or see the human in the experience phase for some duration, particularly before button pressing occurred. Much of this appeared to be due to the subjects already focusing on the button before the human pressed it, or due the subject tracking the human’s hand when he/she began to move it towards the button. Gazing at the human’s face did not seem common. 

We also fitted a model to investigate whether the proportion of time spent gazing at the human changed across trials (see S1 Appendix for details). With increasing trial number, the proportion of time subjects spent looking at the helpful human before and after button pressing increased significantly. This finding may indicate that subjects were learning the importance of the human’s action.

We have now made reference in the general discussion to the possibility that the button and dispenser were more salient than the human and that the dogs did not pay enough attention to the human:

L880 – 885 (General discussion): Finally, it is possible that, as a result of the training, the button and the dispenser were more salient than the human and occupied most of the dogs’ attention throughout both studies. Consequently, the dogs may not have paid enough attention to the human to register fully the relevance of the human’s actions, even though most subjects did at least look at or see the human to some degree. Increasing the salience of the human and the human’s actions could overcome this issue and facilitate the subjects’ registering of the cooperative act.

We have also maintained our discussion from the original version of the manuscript which suggests that natural behaviours may be better to use (L843 – 879).

As far as I understand, you did not perform a preference test between the helpful and the unhelpful human, so you cannot conclude that dogs were not able to discriminate them. Your comparison is indirect and therefore less sensitive. The length of time in which dogs were near each person compared to the owner is not a good indicator of the helper-unhelper discrimination. For example, the presence of the owner during the free interaction session could interfere, decreasing the negative reaction towards the unhelpful human, decreasing the interest in the unfamiliar person, etc. You have to relativize your conclusions in this regard.

We do not agree fully with the argument here. There is no clear reason why the presence of the owner and their potential influence on the dogs’ approach behaviour in the free interaction session would be any different in a direct preference test. Furthermore, a free interaction session like ours has been used successfully in previous studies to observe differences in approach behaviour towards an experimenter at different time points after different experimental treatments. We have nonetheless included a discussion point to address the issue of a free interaction session vs a direct preference test:

L774 – 785: Our assessment of subjects’ preference using the free interaction sessions was admittedly opportunistic and was also indirect, as subjects were never faced with a choice between the helpful and unhelpful human. As a result, our approach is arguably less sensitive than a direct preference test. However, the free interaction session offered the advantage, at least in study 2, of observing subjects interacting with the respective human at a standardized time point after the experience phase and test phase. Had a direct preference test been conducted, the experience with one human would have been more recent than the experience with the other, potentially biasing subjects’ choice. Moreover, free interaction sessions like ours have been used to observe differential approach behaviour towards an experimenter based on experimental treatments immediately prior in inequity aversion studies (66,99). Thus, it seems likely that our assessment was sensitive enough to detect the subjects’ preference (or lack thereof) for either human.

We have also reworded parts of the discussion slightly in relation to the conclusions based on the free interaction session:

L770: …dogs did not appear to develop a preference for a particular human type….

L771 – 773: …suggesting that either they did not distinguish between the two humans based on helpfulness or they did distinguish between them but developed no preference.

L947 – 949: …and a preference for either human type (helpful or unhelpful) was not evident based on a free interaction session.

It is necessary to include some important information about the characteristics of the sample: the description of the breeds and the level of previous training. 

We have now included tables with subject information for studies 1 and 2 (see Table 1 and Table 2): 

Table 1. Subject information including age, sex, breed, and the number of training sessions required to reach the criterion.

Subject ID Age (years) Sex Breed No. of sessions to reach criterion

A1 5 F Australian shepherd 1

B1 10 M Border collie 1

C1 6 M Border collie 1

D1 11 F Border collie 1

E1 8 F Border collie 1

F1 7 M Greyhound 3

G1 5 F Terrier mix 1

H1 11 M Airedale terrier 3

I1 4 M Hungarian vizsla 3

J1 5 M Mix 1

K1 5 F Yorkshire terrier 3

L1 6 M Dachshund 1

M1 4 M Spanish galgo 3

N1 4 M Australian cattle dog mix 1

O1 2 M Labrador retriever mix 1

P1 1 F Border collie 1

Q1 7 F Podenco 2

R1 3 F Beagle 2

S1 3 M German hunting terrier mix 2

T1 5 M Australian shepherd mix 1

U1 9 F Australian shepherd 2

F, female; M, male.

Table 2. Subject information including age, sex, breed, and the number of training sessions required to reach the criterion.

Subject ID Age (years) Sex Breed No. of sessions to reach criterion

A1 6 F Australian shepherd 1

B1 10 F Border collie 1

C1 7 F Border collie 1

D1 12 F Border collie 1

E1 8 F Border collie 1

F2 12 M Border collie 1

G2 5 M Border collie 1

H2 8 F Australian shepherd 2

I2 7 F Mix 2

J2 5 M Border collie mix 1

K2 7 F Golden retriever 1

L2 6 M Bernese mountain dog 1

M2 10 M Labradoodle 2

N2 4 F Mix 2

O2 5 M Border collie 1

P2 2 M Border collie 1

Q2 10 M German shepherd 2

R2 6 F Bernese mountain dog 2

S2 7 F Border collie mix 1

T2 3 M Petit Brabançon 1

U2 5 M Shetland sheepdog 1

F, female; M, male.

Which was the reward used? I imagine that the helpful human gave the dog a better reward than the one the dog obtained by itself during training and motivational trials. 

This has now been clarified in the manuscript:

L298 – 300: The food rewards used in the experience phase were dry food pieces. The type of dry food used in the experience phase for a subject was the same as that used in the training and motivational sessions for that subject.

I can see by comparing Fig 6 and 9 that the performance of the dogs decreased between the two studies. Can you explain this difference? 

It is not clear why this difference exists. We have now included a discussion point on this in the discussion for study 2:

L762 – 767: Regarding overall performance in the test phase, although we did not compare the results of the two studies statistically, median performance in terms of the number of times subjects pressed the button appeared lower in study 2 than in study 1 (see Fig 6 and Fig 9). The reason for this difference is not clear, though it is possible that either the extra training steps in study 2, or having an experience phase and a test phase on the same day, as was the case in study 2, reduced motivation to press in general in the test phase.

I think that the legend of Fig 1 is too long and includes many details that are already included in the text. I suggest reducing it. 

This has now been shortened:

L162 – 165: Fig 1. Layout of test room. Two circles (each approx. 1 m in radius) were marked on the floor of the room, for the free interaction session. A red water bowl was also present in the room. A food dispenser and button are depicted in the middle of the room (note: the dispenser and button were only in this position for the first stage of training). Black curtains surrounding some of the fences are represented by thick black lines.

Please describe the interval between phases. 

It is not clear to which part of the manuscript this refers specifically; however, we have added a description of the break between motivational sessions and test conditions in study 1 and of the break between the experience phase and test phase of study 2:

Description of break between motivational sessions and test conditions (study 1):

L381 – 383: During these breaks, the experimenter changed the setup as necessary and the dog was free to explore the room and drink water.

Description of break between the experience phase and test phase (study 2):

L615 – 617: In between the experience phase and the test phase, subjects were given an approximately two-minute break while the setup was changed. Subjects were free to roam around the room and drink water in this time.

L 324-325 I do not understand the objective of modifying the first trial procedure. 

This has now been clarified in the manuscript:

L320 – 322: Each human began with the opposite button to which they were supposed to press, to facilitate the subject’s understanding that the human intentionally provided or intentionally did not provide food.

L416 “The video for one of the free interaction sessions was unavailable” Do you mean the video from one dog or the videos of all the dogs during one session? In this case, which was the session? 

This has now been clarified in the manuscript:

L413 – 414: A single video recording of one free interaction session for one subject was unavailable due to a technical malfunction.

426 and 551 vs 427. Did you use the number of times or the proportion of times?

We used the proportion and plotted the number of times. This has been amended in the manuscript and supporting information.

Reviewer #2: 

General comments 

In the present study authors tested if dogs would reciprocate a pro-social act from an unknown human in a context involving food, using a novel experimental design adapted from previous studies in dogs and other species. In the first study the authors did not find any evidence of reciprocity in dogs and speculate that this result could have been due to the complexity of the task or the mnemonic demands imposed to the dogs by the time gap between the training and testing phases. Thus, in a second study the authors simplify the task and run the training and testing phases in the same day. Again, dogs failed to reciprocate the human pro social behavior in this simplified version of the task. These findings led the authors to conclude that dogs do not spontaneously reciprocate the human pro-social behavior in contexts involving food. I want to congratulate the authors for their work; I found the paper very interesting and carefully design. I will suggest some minor revisions that could probably improve the paper. 

Specific comments 

Introduction 

The review of the previous literature is precise updated and well written. Nevertheless: 

L. 43-44. I would remove the phrase. “Reliable models of reciprocity that can be manipulated experimentally are scarce”. There is a vast literature on reciprocity on rats, mice, and insects. And I believe that there are all valuable models for the study of reciprocity (e.g. Taborsky’s works).

This has now been removed:

L40 – 42: Many aspects of reciprocity remain poorly understood, however, particularly the proximate mechanisms.

L. 89 -102. I found this paragraph a bit unnecessary. Although, as mention, all the listed characteristics made the dog a good model for the study of pro-sociality in general these are not specific characteristics needed for reciprocity. I would focus in reciprocity (See. Trivers, 1971). 

This paragraph has now been removed and more focus has been placed on characteristics needed for reciprocity:

L88 – 93: Dogs possess additional characteristics which might suggest the propensity to reciprocate help received from humans. Apart from a long history of dog-human cooperation and communication (36–44) for which dogs appear to have evolved complex social cognitive traits (45,46; but see 47), and the development of strong bonds with humans (48–52) which could facilitate reciprocity (53–55), dogs seem to distinguish between cooperative and uncooperative humans.

L118 – 122: Given dogs’ long history of cooperating with humans, their specialized skills for such interaction with humans, their capacity to develop bonds with humans, their ability to discriminate between a cooperative and an uncooperative human, and their propensity to reciprocate help received from conspecifics, we investigated whether pet dogs reciprocate help received from humans.

L. 160-161. In line with the previous comment. Some of the aspects that would enhance direct reciprocity are: the possibility of future encounters (Trivers, 1971), and the bond between the subjects (i.e. De Wal, 1997; Freidin et al., 2013). So I think that one big difference with some previous studies (Quervel-Chaumette et al., 2015; 2016, Sanford et al., 2018; Carballo et al., 2020) is the identity of the receiver. Most evidences showing spontaneous pro social behavior in dogs (or the lack of it, Quervel-Chaumette 2016) came from studies with the owners, or some familiar human, as partner. I understand the methodological choice of the authors to use unknown humans as partners which eliminate a lot of uncontrolled variations that could mixed the results. But I believe that the best solution would be to run a third study whit the owners as partners and counterbalance their role (Helper vs. Unhelper) between subjects. The inclusion or non-inclusion of this third study would not influence in my personal decision to suggest the publication of the paper. I simply mention that this would be an interesting aspect to take into account and probably the authors would like to include a further study or mention this aspect in the conclusion. 

Thank you for the suggestion. We have now included a paragraph on the potential use of familiar partners in the general discussion and we will keep this in mind for future studies:

L939 – 946: In future studies, the familiarity of the partners may also need to be considered. Our use of unfamiliar partners prevented results from being confounded by previous experience with those specific individuals. However, familiarity appears to influence prosociality in dogs, at least in experimental tasks with conspecifics (21,22). Thus, the use of unfamiliar partners may have precluded any prosocial food-giving by the dogs. In addition, much reciprocity in non-human animals is thought to occur within long-term affiliative relationships (53). It may, therefore, be advantageous to study reciprocity between dogs and humans by using humans with whom the dogs are already familiar.

We have also made reference to the unfamiliarity of the partner in relation to discrimination between two partners:

L908 – 909: Moreover, discrimination between unfamiliar humans in our studies may have been particularly demanding.

Methods and analysis 

I found the methodology very well design and implemented. I think the data presented by the authors in the data set of supplementary material and the variables analyzed are clear and with a straightforward interpretation. So I found the statistical analyses unnecessarily a bit too complex for the data and the number of factors taken in consideration. But authors may well prefer to keep the presented analyses.

We have kept the analysis as is but will take this comment on board for future publications.

L. 261. Who was the human who eat the food? 

This has now been clarified in the manuscript:

L252 – 254: The human eating the food in this stage was either the experimenter or a human unfamiliar to the dog; they were not one of the humans who would later act as the helpful or unhelpful human.

Discussion 

Again, I found the discussion well written and pertinent. And I mostly agree with the authors’ interpretation of the data. Nevertheless, I would include some references to the role of learning mechanisms in the development of pro social behavior in dogs and in reciprocity in particular. The authors mentions this when they speculate that with further training and switching roles between dogs and humans dogs probably could reciprocate the human but they do not provide and explicit theoretical link. For example, dogs could be focusing in some other aspect of the experimental situation which could overshadow the role of the helper, thus making the actions of the helper more salient could improve dogs performance.

We have now included discussion points on the potential greater salience of the button and dispenser and the potential benefit of increasing the salience of the human partners and their actions:

L880 – 898 (General discussion): Finally, it is possible that, as a result of the training, the button and the dispenser were more salient than the human and occupied most of the dogs’ attention throughout both studies. Consequently, the dogs may not have paid enough attention to the human to register fully the relevance of the human’s actions, even though most subjects did at least look at or see the human to some degree. Increasing the salience of the human and the human’s actions could overcome this issue and facilitate the subjects’ registering of the cooperative act.

In this context, it is conceivable that the limited training and exposure to the action of the humans also diminished the dogs’ likelihood of perceiving or understanding the human’s action. Gfrerer and Taborsky’s (30,35) training procedure, which included providing a partner with food and receiving food from a partner, required approximately 14 to 19 days with two sessions per day for each subject. Moreover, they incorporated the exchange of roles with a partner. The training protocol in experimental studies in which rats were observed to reciprocate was similar, with repeated experience of exchanging roles prior to the experimental procedure (103,104). The drawn out experience of the receipt of food from a partner, and providing food to a partner, over a number of days, combined with the rapid exchange of roles, may have facilitated an understanding of the significance of the partner’s cooperative or non-cooperative behaviour. In contrast, in our design, the subjects had a single experience day with each human, albeit consisting of numerous trials. Extensive training may be required to facilitate subjects’ attentiveness to the crucial aspects of the setup.

L744 – 761 (study 2 - discussion): It is, nonetheless, conceivable that subjects did not pay enough attention to the humans in the experience phase of either study to permit recognition of the action of the human. We investigated this possibility further by determining whether gazing at the human occurred at all. We focused on the experience phase of study 1 due to better video quality and, more specifically, the experience phase with the helpful human, as it is arguably the more important human to be attentive to in order to reciprocate. 

Visual inspection of plotted data representing the proportion of time spent gazing at the helpful human immediately before button pressing (see S9 Fig) and immediately after button pressing (see S10 Fig) reveals that most subjects generally did look at or see the human in the experience phase for some duration, particularly before button pressing occurred. Much of this appeared to be due to the subjects already focusing on the button before the human pressed it, or due the subject tracking the human’s hand when he/she began to move it towards the button. Gazing at the human’s face did not seem common. 

We also fitted a model to investigate whether the proportion of time spent gazing at the human changed across trials (see S1 Appendix for details). With increasing trial number, the proportion of time subjects spent looking at the helpful human before and after button pressing increased significantly. This finding may indicate that subjects were learning the importance of the human’s action.

Furthermore, the discriminations between unknown humans could be demanding for dogs, thus increasing the difference between helper and un-helper could also improve dogs’ performance or the testing could be done with familiar humans. These aspects are related to the factors that could facilitate reciprocity that I mention earlier (familiarity, bonding, repetition of the encounters etc.) and are usually taken into account when discussing pro-social behavior in dogs. 

We have now included discussion points on the demands of discriminating between the two partners and the potential benefit of increasing the difference between them or testing with familiar humans:

L899 – 910: In the context of learning about crucial aspects of the setup, it is worth noting that to reciprocate directly in our setup, dogs would have needed to discriminate between the helpful and unhelpful human. The discrimination learning required may have been too demanding for such a small number of sessions. Visual and olfactory discrimination learning in dogs can take a considerable number of training sessions (97,105,106). In fact, in a discrimination learning task in which dogs had to discriminate between their owner and a familiar person based on visual information from the faces and heads, an average of 6.5 sessions, each lasting ten trials, were required for dogs to successfully discriminate (107). Similarly, dogs were found to require between three and eight sessions of 20 trials each to successfully discriminate between their owner’s and a stranger’s face (108). Moreover, discrimination between unfamiliar humans in our studies may have been particularly demanding. Increasing the perceptual differences between the humans may help with learning the discrimination in future studies.

L939 – 946: In future studies, the familiarity of the partners may also need to be considered. Our use of unfamiliar partners prevented results from being confounded by previous experience with those specific individuals. However, familiarity appears to influence prosociality in dogs, at least in experimental tasks with conspecifics (21,22). Thus, the use of unfamiliar partners may have precluded any prosocial food-giving by the dogs. In addition, much reciprocity in non-human animals is thought to occur within long-term affiliative relationships (53). It may, therefore, be advantageous to study reciprocity between dogs and humans by using humans with whom the dogs are already familiar.

L909 – 910: Increasing the perceptual differences between the humans may help with learning the discrimination in future studies.

We have also pointed out that some studies observed successful discrimination between a cooperative and an uncooperative partner after quite a small number of trials:

L911 – 922: Despite the apparent difficulty of discrimination learning, however, it is worth keeping in mind that dogs were previously shown by Gfrerer and Taborsky (35) to reciprocate after just 14 trials of experience with each partner type. It is plausible that the more extensive training procedure, outlined above, facilitated discrimination by priming subjects to attend to the partner. Furthermore, partner discrimination may be easier if the partners are conspecifics. However, as subjects received an experience phase and test phase with a single partner in a two-day sequence before encountering a new partner (35), discrimination learning may not have been necessary to facilitate reciprocity. Nonetheless, Heberlein et al. (56) and Carballo et al. (57) found that dogs could discriminate between humans that differed in cooperativeness after just six and twelve trials respectively. As subjects in our studies were presented with each human across at least 20 trials, it is not clear that the challenge of discrimination learning can account for the lack of reciprocity.

L. 885: include “direct” before reciprocity.

This has now been included (see L952).

---

## [Decision Letter · Decision Letter 1]

2 Jun 2021

Dogs fail to reciprocate the receipt of food from a human in a food-giving task

PONE-D-20-35864R1

Dear Dr. McGetrick,

We’re pleased to inform you that your manuscript has been judged scientifically suitable for publication and will be formally accepted for publication once it meets all outstanding technical requirements.

Kind regards,

Simon Clegg, PhD

Academic Editor

PLOS ONE

Additional Editor Comments:

Many thanks for resubmitting your manuscript to PLOS One

The reviewers are happy that you have addressed all the comments and the manuscript reads well, therefore I have recommended it for publication

You should hear from the Editorial Office shortly.

It was a pleasure working with you and I wish you the best of luck for your future research

Hope you are keeping safe and well in these difficult times

Thanks

Simon

Reviewers' comments:

Reviewer's Responses to Questions

**Comments to the Author**

1. If the authors have adequately addressed your comments raised in a previous round of review and you feel that this manuscript is now acceptable for publication, you may indicate that here to bypass the “Comments to the Author” section, enter your conflict of interest statement in the “Confidential to Editor” section, and submit your "Accept" recommendation.

Reviewer #1: All comments have been addressed

Reviewer #2: All comments have been addressed

2. Is the manuscript technically sound, and do the data support the conclusions?

Reviewer #1: (No Response)

Reviewer #2: Yes

3. Has the statistical analysis been performed appropriately and rigorously? 

Reviewer #1: (No Response)

Reviewer #2: Yes

4. Have the authors made all data underlying the findings in their manuscript fully available?

Reviewer #1: (No Response)

Reviewer #2: Yes

5. Is the manuscript presented in an intelligible fashion and written in standard English?

Reviewer #1: (No Response)

Reviewer #2: Yes

6. Review Comments to the Author

Reviewer #1: (No Response)

Reviewer #2: I found all my comments responded and congratulate you for your work.

One final comment. In the figures 6 and 9 you represent the number of times the subject press the button while in the analysis and results sections you refered to the proportion fo times dogs press the button. For consistency I would change the figure's y axis

7. PLOS authors have the option to publish the peer review history of their article (what does this mean?). If published, this will include your full peer review and any attached files.

Reviewer #1: No

---

## [Editor Report · Acceptance letter]

21 Jun 2021

PONE-D-20-35864R1 

Dogs fail to reciprocate the receipt of food from a human in a food-giving task 

Dear Dr. McGetrick:

I'm pleased to inform you that your manuscript has been deemed suitable for publication in PLOS ONE. Congratulations! Your manuscript is now with our production department. 

Kind regards, 

on behalf of

Dr. Simon Clegg 

Academic Editor

PLOS ONE